# eIF1A residues implicated in cancer stabilize translation preinitiation complexes and favor suboptimal initiation sites in yeast

Pilar Martin-Marcos[1,2], Fujun Zhou[3], Charm Karunasiri[3], Fan Zhang[1], Jinsheng Dong[1], Jagpreet Nanda[3], Shardul D Kulkarni [3], Neelam Dabas Sen[1], Mercedes Tamame[2], Michael Zeschnigk[4,5], Jon R Lorsch[3]*, Alan G Hinnebusch[1]*

[1]Laboratory of Gene Regulation and Development, Eunice Kennedy Shriver National Institute of Child Health and Development, National Institutes of Health, Bethesda, United States; [2]Instituto de Biología Funcional y Genómica, IBFG-CSIC, Universidad de Salamanca, Salamanca, Spain; [3]Laboratory on the Mechanism and Regulation of Protein Synthesis, Eunice Kennedy Shriver National Institute of Child Health and Development, National Institutes of Health, Bethesda, United States; [4]Institute of Human Genetics, University Duisburg-Essen, Essen, Germany; [5]Eye Cancer Research Group, Faculty of Medicine, University Duisburg-Essen, Essen, Germany

**Abstract** The translation pre-initiation complex (PIC) scans the mRNA for an AUG codon in favorable context, and AUG recognition stabilizes a closed PIC conformation. The unstructured N-terminal tail (NTT) of yeast eIF1A deploys five basic residues to contact tRNA$_i$, mRNA, or 18S rRNA exclusively in the closed state. Interestingly, *EIF1AX* mutations altering the human eIF1A NTT are associated with uveal melanoma (UM). We found that substituting all five basic residues, and seven UM-associated substitutions, in yeast eIF1A suppresses initiation at near-cognate UUG codons and AUGs in poor context. Ribosome profiling of NTT substitution R13P reveals heightened discrimination against unfavorable AUG context genome-wide. Both R13P and K16D substitutions destabilize the closed complex at UUG codons in reconstituted PICs. Thus, electrostatic interactions involving the eIF1A NTT stabilize the closed conformation and promote utilization of suboptimal start codons. We predict UM-associated mutations alter human gene expression by increasing discrimination against poor initiation sites.

DOI: https://doi.org/10.7554/eLife.31250.001

*For correspondence:
jon.lorsch@nih.gov (JRL);
ahinnebusch@nih.gov (AGH)

## Introduction

Accurate identification of the translation initiation codon in mRNA by ribosomes is crucial for expression of the correct cellular proteins. This process generally occurs in eukaryotic cells by a scanning mechanism, wherein the small (40S) ribosomal subunit recruits charged initiator tRNA (Met-tRNA$_i^{Met}$) in a ternary complex (TC) with eIF2-GTP, and the resulting 43S pre-initiation complex (PIC) attaches to the 5' end of the mRNA and scans the 5'UTR for an AUG start codon. In the scanning PIC, the TC is bound in a relatively unstable state, dubbed 'P$_{OUT}$', suitable for inspecting successive triplets in the P decoding site for perfect complementarity with the anticodon of Met-tRNA$_i$. Nucleotides surrounding the AUG, particularly at the −3 and +4 positions (relative to the AUG at +1 to +3), the 'Kozak context', also influence the efficiency of start codon recognition. Hydrolysis of the GTP bound to eIF2 can occur, dependent on GTPase activating protein eIF5, but P$_i$ release is blocked by eIF1, whose presence also prevents highly stable binding of Met-tRNA$_i^{Met}$ in the 'P$_{IN}$' state. Start-codon

recognition triggers dissociation of eIF1 from the 40S subunit, which in concert with other events allows $P_i$ release from eIF2-GDP·$P_i$ and accommodation of Met-tRNA$_i^{Met}$ in the $P_{IN}$ state of the 48S PIC (*Figure 1A*). Subsequent dissociation of eIF2-GDP and other eIFs from the 48S PIC enables eIF5B-catalyzed subunit joining and formation of an 80S initiation complex with Met-tRNA$_i^{Met}$ base-paired to AUG in the P site (reviewed in [*Hinnebusch, 2014*] and [*Hinnebusch, 2017*]). eIF1 plays a dual role in the scanning mechanism, promoting rapid TC loading in the $P_{OUT}$ conformation while blocking rearrangement to $P_{IN}$ at non-AUG codons by clashing with Met-tRNA$_i$ in the $P_{IN}$ state (*Rabl et al., 2011*; *Lomakin and Steitz, 2013*)(*Hussain et al., 2014*), thus requiring dissociation of eIF1 from the 40S subunit for start codon recognition (*Figure 1A*). Consequently, mutations that weaken eIF1 binding to the 40S subunit reduce the rate of TC loading and elevate initiation at near-cognate codons (eg. UUG), or AUG codons in poor context, by destabilizing the open/$P_{OUT}$ conformation and favoring rearrangement to the closed/$P_{IN}$ state during scanning (*Martin-Marcos et al., 2011*; *Martin-Marcos et al., 2013*). Moreover, decreasing wild-type (WT) eIF1 abundance reduces initiation accuracy, whereas overexpressing eIF1 suppresses initiation at near-cognates or AUGs in poor context (*Valásek et al., 2004*; *Alone et al., 2008*; *Ivanov et al., 2010*; *Saini et al., 2010*; *Martin-Marcos et al., 2011*). The mechanistic link between eIF1 abundance and initiation accuracy is exploited to negatively autoregulate eIF1 expression, as the AUG codon of the eIF1 gene (*SUI1* in yeast) occurs in suboptimal context and the frequency of its recognition is inversely related to eIF1 abundance (*Ivanov et al., 2010*; *Martin-Marcos et al., 2011*). Mutations that weaken 40S binding by eIF1 relax discrimination against the poor context of the *SUI1* AUG codon and elevate eIF1 expression, overcoming autoregulation (*Martin-Marcos et al., 2011*). In contrast, mutations that enhance eIF1 binding to the 40S subunit impede rearrangement of the scanning PIC to the closed/$P_{IN}$ conformation (*Martin-Marcos et al., 2011*; *Martin-Marcos et al., 2014*), which increases discrimination against the poor context of the *SUI1* AUG codon, to reduce eIF1 expression, and also suppresses initiation at near-cognate UUG codons (*Martin-Marcos et al., 2011*; *Martin-Marcos et al., 2014*).

eIF1A also has a dual role in scanning and start codon recognition. Scanning enhancer (SE) elements in the eIF1A C-terminal tail (CTT) promote TC binding in the open $P_{OUT}$ conformation and impede rearrangement to the closed $P_{IN}$ state, such that substitutions that impair the SE elements both impair TC recruitment and increase initiation at near-cognate start codons (*Saini et al., 2010*). Biochemical mapping experiments suggest that, like eIF1, the eIF1A CTT clashes with Met-tRNA$_i$ in the $P_{IN}$ state (*Yu et al., 2009*), and is displaced from the P site on start codon recognition (*Zhang et al., 2015*) to enable a functional interaction of the eIF1A CTT with the NTD of eIF5, the GTPase activating protein for eIF2, that facilitates $P_i$ release from eIF2-GDP·$P_i$ (*Nanda et al., 2013*). Scanning inhibitor elements $SI_1$ and $SI_2$ in the unstructured eIF1A N-terminal tail (NTT) and helical domain, respectively, antagonize SE function and stabilize the closed/$P_{IN}$ conformation on start codon recognition (*Figure 1A*). Accordingly, substitutions that impair SI elements destabilize the closed complex and accelerate TC loading to the open complex in vitro, and promote continued scanning at UUG codons in hypoaccurate mutant cells (*Fekete et al., 2007*) (*Saini et al., 2010*). $SI_1$ mutations also increase the probability that the scanning PIC will bypass an upstream AUG codon (leaky scanning) (*Fekete et al., 2007*; *Luna et al., 2013*); and one such mutation, substituting NTT residues 17–21, decreases recognition of the suboptimal AUG codon of *SUI1* mRNA to reduce eIF1 expression (*Martin-Marcos et al., 2011*).

Molecular insight into the deduced function of the eIF1A-NTT of promoting AUG recognition during scanning came from the cryo-EM structure of a partial yeast 48S PIC (py48S) containing eIF1, eIF1A, TC and mRNA, with the Met-tRNA$_i$ base-paired to the AUG codon in a $P_{IN}$ state. All but the first four residues of the eIF1A NTT were visible in this structure, and basic NTT residues Lys7, Lys10, Arg13, and Lys16 contact either the anticodon or the +4 to +6 mRNA nucleotides adjacent to the AUG codon, while Arg14 interacts with the 18S rRNA (*Figure 1B*) (*Hussain et al., 2014*). These findings suggest that the eIF1A NTT can directly stabilize the $P_{IN}$ state, and help to explain how NTT substitutions in $SI_1$, which spans residues 1–26 (*Saini et al., 2010*), increase discrimination against non-AUG codons, which form less stable codon:anticodon duplexes than do AUG codons. Other studies have implicated NTT residues 7–11 and 12–16, encompassing the aforementioned basic NTT residues, in interactions with eIF1 and the eIF5-CTD that appear to promote assembly of the open, scanning PIC (*Fekete et al., 2007*; *Luna et al., 2013*). The β-subunit of eIF2 also harbors a highly basic NTT, which interacts with the eIF5-CTD to promote eIF1 dissociation from the closed complex

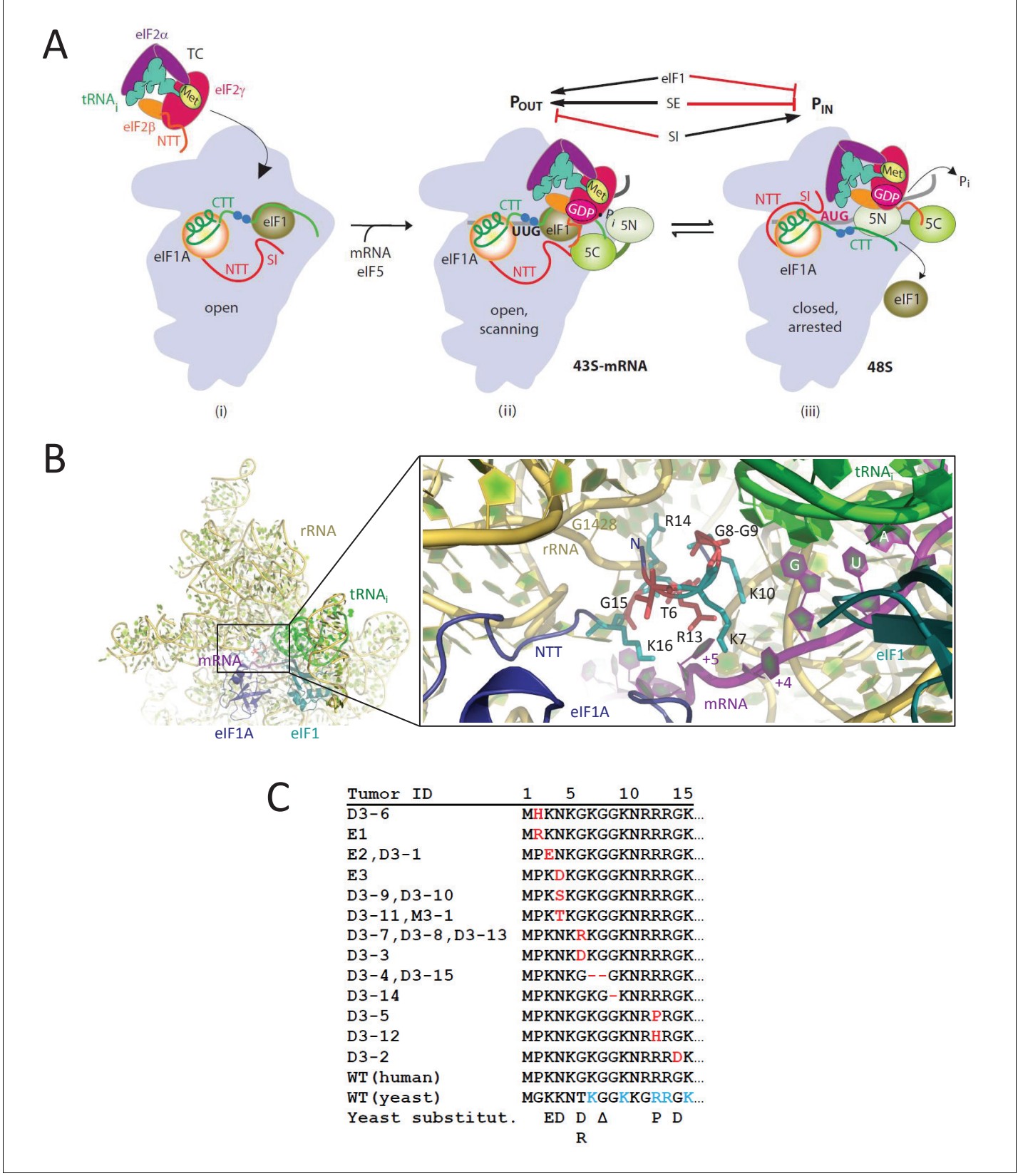

**Figure 1.** eIF1A-NTT residues associated with UM are predicted to participate in stabilizing the $P_{IN}$ state of the closed conformation of the 48S PIC. (**A**) Model describing known conformational rearrangements of the PIC during scanning and start codon recognition. (i) eIF1 and the scanning enhancers

*Figure 1 continued on next page*

*Figure 1 continued*

(blue balls) in the C-terminal tail (CTT) of eIF1A stabilize an open conformation of the 40S subunit to which TC rapidly binds. (ii) The 43S PIC in the open conformation scans the mRNA for the start codon with Met-tRNA$_i$$^{Met}$ bound in the P$_{OUT}$ state. eIF2 can hydrolyze GTP to GDP•P$_i$, but release of P$_i$ is blocked by eIF1. The N-terminal tail (NTT) of eIF1A interacts with the eIF5-CTD. (iii) On AUG recognition, Met-tRNA$_i$$^{Met}$ moves from the P$_{OUT}$ to P$_{IN}$ state, clashing with eIF1 and the CTT of eIF1A, provoking displacement of the eIF1A CTT from the P site, dissociation of eIF1 from the 40S subunit, and P$_i$ release from eIF2. The NTT of eIF2β interacts with the eIF5-CTD, and the eIF1A-NTT, harboring scanning inhibitor (SI) elements, interacts with the codon:anticodon helix. (Above) Arrows summarize that eIF1 and the eIF1A SE elements promote P$_{OUT}$ and impede transition to P$_{IN}$ state, whereas the eIF1A SI element in the NTT stabilizes the P$_{IN}$ state. (Adapted from (*Hinnebusch, 2014*)). Results presented below show that this function of the eIF1A-NTT is impaired by uveal melanoma (UM)-associated substitutions and others that disrupt direct contacts with the mRNA or codon:anticodon helix shown in (B). (B) Magnified portion of the py48S PIC structure (PDB 3J81) showing contacts made by the eIF1A-NTT (shades of blue and cyan) in the closed/P$_{IN}$ conformation. Side-chains of NTT residues substituted in UM (red) or directly contacting 18S rRNA (yellow), tRNA$_i$ (green) or mRNA (purple) are shown as sticks. (C) Sequence of human eIF1A NTT residues 1–15 showing the substitutions (red) or deletions (dash) found in the indicated UM tumors. Substitutions in yeast eIF1A corresponding to those found in UM tumors are listed on the last line. The five basic residues of the yeast NTT making direct contacts in the PIC and substituted here in addition to the UM-associated substitutions are shown in cyan.

DOI: https://doi.org/10.7554/eLife.31250.002

at the start codon (*Luna et al., 2012*). It was suggested that interaction of the eIF5-CTD with the eIF1A-NTT would stabilize the open conformation of the PIC prior to AUG recognition, whereas alternative interaction of the eIF5-CTD with the eIF2β-NTT would stabilize the closed conformation of the PIC on AUG recognition (*Luna et al., 2013*). The proposed dissociation of the eIF1A-NTT from the eIF5-CTD on AUG recognition would free the eIF1A-NTT for interactions with the mRNA and anticodon evident in the py48S PIC (*Hussain et al., 2014*). Thus, the eIF1A-NTT would play a dual role of promoting the open conformation of the PIC through interaction with the eIF5-CTD and subsequently stabilizing the closed state by interacting with the mRNA and anticodon.

Somatic mutations in the human gene *EIF1AX* encoding eIF1A are frequently associated with uveal melanomas (UM) associated with disomy for chromosome 3, and all of the *EIF1AX* mutations sequenced thus far produce in-frame substitutions or deletions of one or more residues in the first 15 residues of the NTT (*Martin et al., 2013*). A subset of these mutations substitute or delete two of the four basic residues that contact mRNA or the tRNA$_i$ anticodon in the yeast py48S PIC (Lys7 and Arg13), others introduce acidic residues that might electrostatically repel the phosphodiester backbone of the mRNA or tRNA$_i$, while others affect a Gly-Gly turn that is important for correct positioning of the basic residues in the PIC (*Figure 1B–C*) (*Hussain et al., 2014*). Thus, all of the UM mutations might affect eIF1A function by the same mechanism, of weakening the ability of the eIF1A NTT to stabilize the P$_{IN}$ conformation of the tRNA$_i$. As such, they would be expected to increase discrimination against poor initiation sites in vivo. Alternatively, they could impair the proposed function of the eIF1A-NTT in stabilizing the open conformation (*Luna et al., 2013*), in which case they would have the opposite effect and relax discrimination against suboptimal start codons. We set out to distinguish between these possibilities by examining the consequences of seven yeast eIF1A-NTT substitutions equivalent to those associated with UM in residues Lys3, Lys4, Thr6, Gly8, Arg13 and Gly15, and also of altering the five NTT basic residues that interact with the mRNA or anticodon in the py48S PIC (Lys7, Lys10, Arg13, Arg14 and Lys16) (*Figure 1C*). Our genetic and biochemical analyses indicate that UM-associated eIF1A substitutions disrupt NTT interactions with the mRNA or tRNA$_i$ to destabilize the closed/P$_{IN}$ conformation of the PIC and increase discrimination against near-cognate codons or AUGs in suboptimal context, with particularly strong effects observed for substitutions of Arg13—one of five basic residues that interacts with the mRNA/tRNA$_i$ anticodon. Ribosome profiling of the potent UM-associated mutant eIF1A-R13P reveals widespread increased discrimination against AUG codons in poor context, which can alter recognition of the start codon for the main coding sequences (CDS) or indirectly affect translation by modulating recognition of upstream open reading frames (uORFs) in the mRNA leader. These findings allow us to predict that eIF1A-NTT mutations alter gene expression in UM tumor cells by shifting translation initiation at main CDS and regulatory uORFs from poor to optimum initiation sites.

## Results

### UM-associated substitutions in the yeast eIF1A NTT increase discrimination against near-cognate UUG codons in vivo

To explore functional consequences of substitutions in human eIF1A associated with uveal melanoma (*Martin et al., 2013*), we introduced substitutions into the yeast eIF1A NTT corresponding to 7 of the 13 substitutions associated with the disease: K3D, K4D, T6R, T6D, ΔG8, R13P, and G15D (*Figure 1C*). Asn4 and Gly6 of human eIF1A correspond to Lys4 and Thr6 in yeast, thus the yeast K4D and T6R/T6D substitutions mimic the human N4D and G6R/G6D UM-associated substitutions, respectively. The deletion of Gly8 (ΔG8) in yeast produces the same protein as the UM-associated substitution ΔG9, leaving a single Gly residue in place of the Gly8/Gly9 pair (*Figure 1C*). Mutations were generated in a *TIF11* allele (encoding yeast eIF1A) under the native promoter and the mutant alleles on single-copy plasmids were used to replace WT *TIF11⁺* by plasmid-shuffling in a *his4-301* yeast strain in order to examine their effects on initiation at near-cognate UUG codons. *his4-301* lacks an AUG start codon and confers histidine auxotropy, which can be suppressed by mutations in initiation factors with the Sui⁻ phenotype (Suppressor of initiation codon mutation) that increase initiation at the third, in-frame UUG codon to restore expression of histidine biosynthetic enzyme His4 (*Donahue, 2000*). None of the *TIF11* mutations allows detectable growth on medium containing only 1% of the usual histidine supplement (*Figure 2—figure supplement 1A*, -His medium), indicating the absence of Sui⁻ phenotypes; and none confers a slow-growth phenotype (Slg⁻) on complete medium (*Figure 2—figure supplement 1A*, +His). We next tested the mutant alleles for the ability to suppress the elevated UUG initiation on *his4-301* mRNA and the attendant His⁺ phenotype conferred by dominant Sui⁻ mutations *SUI5* and *SUI3-2* encoding, respectively, the G31R variant of eIF5 and S264Y variant of eIF2β (*Huang et al., 1997*). Remarkably, the dominant His⁺ phenotypes conferred by plasmid-borne *SUI5* or *SUI3-2* are diminished by all of the NTT mutations (*Figure 2A* and *Figure 2—figure supplement 1B*, -His); and the Slg⁻ phenotype conferred by *SUI5* in +His medium at 37°C is also suppressed by the *K3E, K4D, ΔG8, R13P,* and *G15D* mutations (*Figure 2A*, +His, 37°C). These results suggest that the UM-associated substitutions, as a group, mitigate the effects of *SUI5* and *SUI3-2* in elevating UUG initiation, increasing discrimination against near-cognate start codons.

The effect of *SUI3-2* in reducing the fidelity of start codon selection can be quantified by measuring the expression of matched *HIS4-lacZ* reporters containing a UUG or AUG triplet as start codon. As expected (*Huang et al., 1997*), *SUI3-2* increases the ratio of expression of the UUG to AUG reporter from the low WT value of ~3% up to ~12% (*Figure 2B*). With the exception of *T6D*, all of the UM mutations significantly reduced the *HIS4-lacZ* UUG:AUG initiation ratio, with *R13P* eliminating ~75% of the increase in the UUG/AUG initiation ratio conferred by *SUI3-2* in *TIF11⁺* cells (*Figure 2B*). The results indicate that eIF1A UM substitutions restore to varying extents discrimination against near-cognate UUG codons in Sui⁻ mutants, thus conferring Ssu⁻ phenotypes.

Many Sui⁻ mutations, including *SUI3-2*, derepress *GCN4* mRNA translation in nutrient-replete cells (the Gcd⁻ phenotype) (*Saini et al., 2010*). This phenotype generally results from a reduced rate of TC recruitment that allows 40S subunits that have translated upstream open reading frame 1 (uORF1) and resumed scanning to subsequently bypass uORFs 2–4 and reinitiate at the *GCN4* AUG codon in the absence of a starvation-induced inhibition of TC assembly (*Hinnebusch, 2005*). Interestingly, the Gcd⁻ phenotype of *SUI3-2*, manifested as an ~3 fold derepression of a *GCN4-lacZ* reporter, is also significantly diminished by *R13P* (*Figure 2—figure supplement 1C*), the eIF1A NTT mutation shown above to be the strongest suppressor of the Sui⁻ phenotype of *SUI3-2* (*Figure 2B*). Co-suppression of the Gcd⁻ and Sui⁻ phenotypes of *SUI3-2* has been demonstrated for other Ssu⁻ mutations in eIF1A (*Saini et al., 2010*; *Dong et al., 2014*; *Martin-Marcos et al., 2014*) and attributed to destabilization of the closed/P$_{IN}$ conformation and attendant shift to the open scanning-conducive conformation to which TC binds rapidly (*Passmore et al., 2007*). Thus, co-suppression of the Gcd⁻ and Sui⁻/hypoaccuracy phenotypes of *SUI3-2* observed only for the *R13P* mutation suggests that it exceeds the other UM-associated mutations in destabilizing the closed/P$_{IN}$ conformation of the PIC.

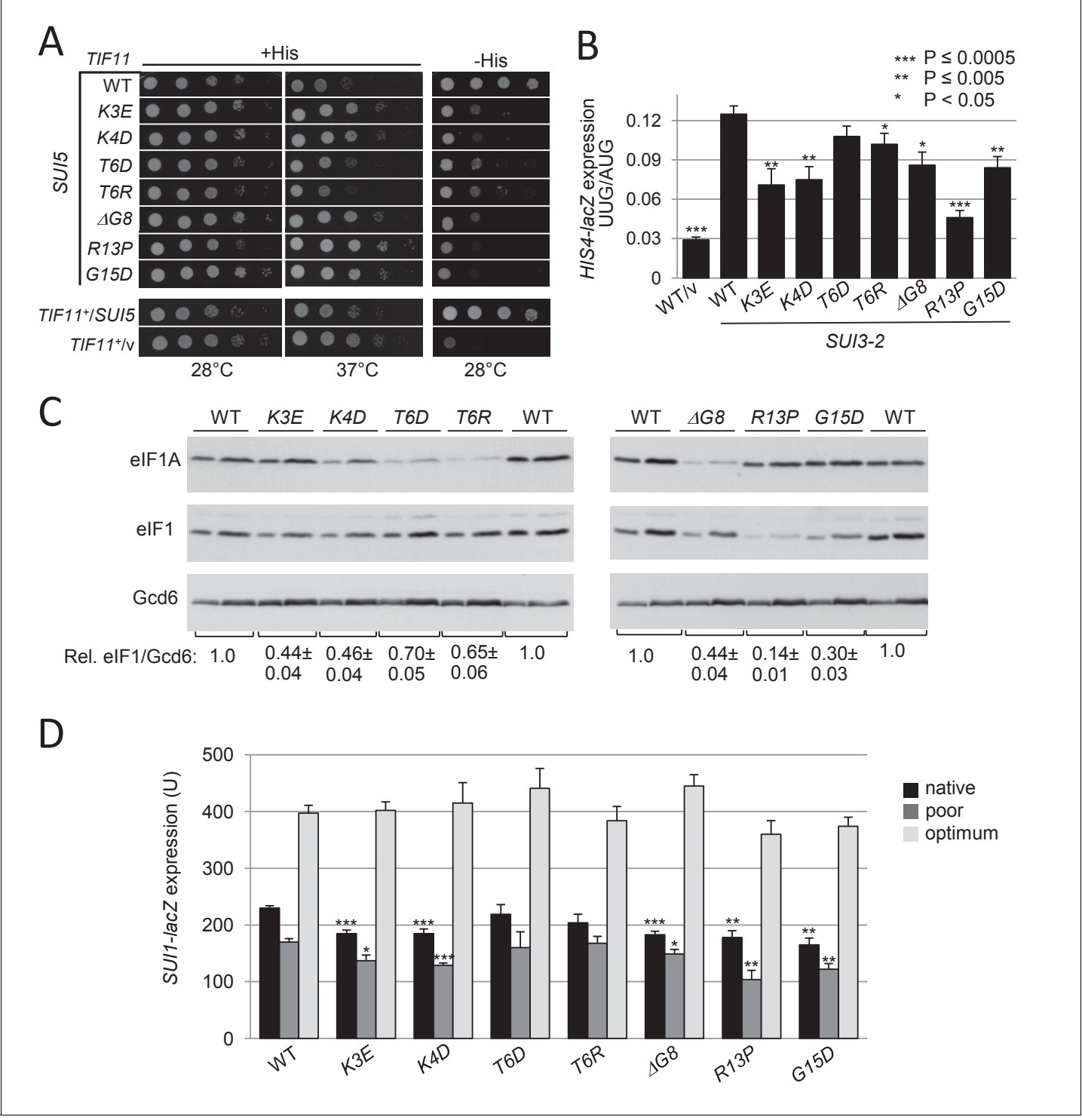

**Figure 2.** UM-associated substitutions in the yeast eIF1A NTT suppress Sui⁻ phenotypes conferred by mutations *SUI5* and *SUI3-2* and increase discrimination against the poor, native start codon of *SUI1* mRNA. (**A**) Ten-fold serial dilutions of *tif11Δ his4-301* strain H3582 containing the indicated *TIF11* (eIF1A) alleles on single copy (sc) plasmids and either episomal *SUI5* (p4281/YCpTIF5-G31R-W) or empty vector (/v) were analyzed for Slg⁻ and His⁺/Sui⁻ phenotypes on SC lacking leucine (Leu) and tryptophan (Trp) supplemented with 0.3 mM His and incubated at 28°C or 37°C for 2 days (+His), or on SC-Leu-Trp plus 0.003 mM His (-His) and grown at 28°C for 4 days. (**B**) Derivatives of strain H3582 containing the indicated *TIF11* alleles and episomal *SUI3-2* (p4280/YCpSUI3-S264Y-W) or empty vector (/v) and also harboring *HIS4-lacZ* reporters with AUG or UUG start codons (plasmids p367 and p391, respectively) were cultured in synthetic dextrose minimal medium (SD) supplemented with His at 28°C to A₆₀₀ of ~1.0, and β-galactosidase activities (in units of nanomoles of *o*-nitrophenyl-β-D-galactopyranoside cleaved per min per mg) were measured in whole cell extracts (WCEs). The ratio

*Figure 2 continued on next page*

*Figure 2 continued*

of expression of the UUG to AUG reporter was calculated from at least four different measurements, and the mean and S.E.M.s were plotted. (**C**) Derivatives of H3582 containing the indicated *TIF11* alleles were cultured in SD supplemented with His, Trp and uracil (Ura) at 28°C to $A_{600}$ of ~1.0, and WCEs were subjected to Western analysis using antibodies against eIF1A/Tif11, eIF1/Sui1 or eIF2Bε/Gcd6 (analyzed as loading control). Two different amounts of each extract differing by 2-fold were loaded in successive lanes. (**D**) Same strains as in (**C**) harboring the sc plasmids with *SUI1-lacZ* fusions containing the native suboptimal ($_{-3}$CGU$_{-1}$, pPMB24), poor ($_{-3}$UUU$_{-1}$, pPMB28) or optimum ($_{-3}$AAA$_{-1}$, pPMB25) AUG contexts were cultured in SD +His + Trp at 28°C to $A_{600}$ of ~1.0, and assayed for β-galactosidase activities as in (**B**).

DOI: https://doi.org/10.7554/eLife.31250.003

The following source data and figure supplement are available for figure 2:

**Source data 1.** Source data for *Figure 2* and *Figure 2—figure supplement 1*.
DOI: https://doi.org/10.7554/eLife.31250.005
**Figure supplement 1.** UM-associated eIF1A NTT substitutions reduce the His$^+$/Sui$^-$ and Gcd$^-$ phenotypes of *SUI3-2*.
DOI: https://doi.org/10.7554/eLife.31250.004

## UM-associated eIF1A substitutions increase discrimination against AUG codons in poor context

In addition to reducing initiation at near-cognate UUG codons in Sui$^-$ mutants, Ssu$^-$ substitutions in eIF1 and eIF1A are known to increase discrimination against the AUG start codon of the *SUI1* gene encoding eIF1, which exhibits a non-preferred Kozak context. The unfavorable context of the *SUI1* start codon underlies negative autoregulation of eIF1 synthesis, which hinders overexpression of eIF1 as excess eIF1 impedes initiation at its own start codon (*Martin-Marcos et al., 2011*). Consistent with this, the eIF1A UM mutations reduce the steady-state level of eIF1, with the strongest reduction for *R13P*, lesser reductions for *K3E, K4D, ΔG8*, and *G15D*, and the smallest effects for *T6R* and *T6D* (*Figure 2C*, eIF1 blot and eIF1/Gcd6 ratios). This hierarchy exactly parallels that observed for suppression of the UUG:AUG initiation ratio in *SUI3-2* cells for these eIF1A mutants (*Figure 2B*).

Results in *Figure 2C* also reveal that *K4D, ΔG8, T6R* and *T6D* reduce expression of eIF1A itself (eIF1A blot). It seems unlikely that these reductions arise from altered translation of eIF1A, as the eIF1A AUG codon is in good context (A at −3) and the reductions do not correlate with decreases in eIF1 expression conferred by different eIF1A variants (*Figure 2C*). Rather, these substitutions, and those at Lys10 discussed below (Figure 4A), might impair a role of the first 10 residues of eIF1A in stabilizing the protein. Regardless, the reduced expression of these eIF1A variants is insufficient to confer a marked reduction in eIF1 synthesis or a strong Ssu$^-$ phenotype, as both eIF1A-T6R and eIF1A-T6D are poorly expressed but have a small impact on both eIF1 expression (*Figure 2C*) and the enhanced UUG initiation conferred by *SUI3-2* (*Figure 2B*). Furthermore, we show below that increasing the expression of the eIF1A-K4D and -ΔG8 variants does not diminish their effects on UUG initiation or eIF1 expression.

In accordance with their effects on eIF1 expression, the *R13P, K3E, K4D, ΔG8,* and *G15D* mutations significantly reduce expression of the WT *SUI1-lacZ* fusion containing the native, poor context of the eIF1 AUG codon, $_{-3}$CGU$_{-1}$ (*Figure 2D*, Native context). These eIF1A mutations also reduce expression of a second reporter in which the native AUG context is replaced with the even less favorable context of $_{-3}$UUU$_{-1}$ (*Martin-Marcos et al., 2011*), with *R13P* again conferring the largest reduction (*Figure 2D*, poor context). Importantly, none of the mutations significantly affects expression of a third reporter containing the optimum context of $_{-3}$AAA$_{-1}$ (*Martin-Marcos et al., 2011*) (*Figure 2D*). Thus, a subset of the UM mutations, and particularly *R13P* and *G15D*, selectively reduce recognition of the eIF1 AUG codon when it resides in its native poor context, or in another poor context, in addition to increasing discrimination against near-cognate UUG start codons.

## NTT basic residues contacting mRNA or tRNA$_i$ in the py48S complex have a role in recognition of poor initiation sites in vivo

Among the UM mutations, *R13P* consistently conferred the greatest reduction in recognition of both UUG codons and AUGs in poor context (*Figure 2* and *Figure 2—figure supplement 1*). In the structure of py48S, Arg13 contacts the +5 nucleotide in mRNA, and with Lys7, Lys10, and Lys16, is one of four basic residues in the eIF1A NTT contacting the mRNA or tRNA$_i$ anticodon (*Figure 1B*). A fifth basic residue, Arg14 contacts A1427/G1428 of 18S rRNA located in the mRNA binding cleft (*Hussain et al., 2014*). In addition, UM mutation ΔG8 affects the tandem Gly8-Gly9 pair that

mediates a turn in the NTT required for proper positioning of the four basic residues. Accordingly, we hypothesized that the hyperaccuracy phenotypes of the UM-associated substitutions R13P and ΔG8 reflect loss of a direct contact with the mRNA (R13P) or perturbation of one or more contacts of the four basic residues with mRNA/tRNA$_i$ (ΔG8), which destabilizes the P$_{IN}$ state of the 48S PIC. Moreover, insertion of an acidic side-chain between basic residues Arg14 and Lys16 by UM substitution G15D (*Figure 1B*), which could introduce electrostatic repulsion with the backbone of mRNA or rRNA, could likewise destabilize the 48S PIC. Because UUG start codons form a less stable codon:anticodon helix with a U:U mismatch compared to the perfect duplex formed at AUG codons, UM substitutions that destabilize P$_{IN}$ should be especially deleterious to initiation at UUG codons, as we observed (*Figure 2*). To test this hypothesis, we introduced Ala and Asp substitutions at all five of the NTT basic residues that contact mRNA, tRNA$_i$ or rRNA in the py48S PIC (*Hussain et al., 2014*), expecting to find stronger hyperaccuracy phenotypes for Asp versus Ala substitutions owing to electrostatic repulsion with the nucleic acids in the case of Asp replacements. We also examined a double deletion of Gly8-Gly9 that we reasoned might have a stronger phenotype than the UM mutation ΔG8.

We observed modest Slg$^-$ phenotypes for the R13D and R14D substitutions, but no His$^+$ phenotypes indicative of Sui$^-$ defects for any of the targeted NTT mutations (*Figure 3—figure supplement 1A*). Remarkably, both Ala and Asp substitutions of Lys10, Arg13, Arg14, and Lys16, and the Asp substitution of Lys7, all diminished the His$^+$/Sui$^-$ phenotype of *SUI3-2* (*Figure 3A*) and decreased the *HIS4-lacZ* UUG:AUG initiation ratio in *SUI3-2* cells, with the greatest reductions seen for R13D, R14D, and K16D. In agreement with our hypothesis, the Asp versus Ala substitutions generally conferred greater suppression of the UUG:AUG ratio, but especially so at Lys10 and Lys16 (*Figure 3B*). Using a second set of UUG and AUG reporters, expressing renilla or firefly luciferase from different transcripts under the control of the *ADH1* (*RLUC*) or *GPD* (*FLUC*) promoter, we confirmed that the K16D and R13P substitutions reduced the elevated UUG:AUG initiation ratio conferred by *SUI3-2* (*Figure 3—figure supplement 1B*). All of the mutations, except for K7A, also diminished the Gcd$^-$ phenotype of *SUI3-2*, reducing the derepression of *GCN4-lacZ* expression, again with generally greater reductions for Asp versus Ala replacements (*Figure 3C*). The degree of suppression of the elevated UUG:AUG ratio and *GCN4-lacZ* expression in *SUI3-2* cells was correlated, with *R13D, R14D,* and *K16D* being the strongest suppressors of both phenotypes (cf. *Figure 3B and C*). As noted above, this co-suppression of impaired TC loading (Gcd$^-$) and increased UUG recognition (Sui$^-$) phenotypes suggest that these eIF1A NTT substitutions specifically destabilize the closed/P$_{IN}$ state with attendant shift to the open/P$_{OUT}$ scanning conformation of the PIC (*Saini et al., 2010*).

In addition to suppressing UUG initiation, all of the targeted substitutions of the five basic residues, and the deletion of Gly8-Gly9, also increase discrimination against the non-preferred context of the eIF1 AUG codon, reducing expression of eIF1 (*Figure 4A*) and of the *SUI1-lacZ* fusions with native or poor context, without altering expression of *SUI1-lacZ* with optimal AUG context (*Figure 4B*). Again, the Asp versus Ala substitutions of the basic NTT residues generally confer stronger phenotypes (*Figure 4A–B*), consistent with stronger disruptions of NTT contacts with mRNA, tRNA$_i$ or rRNA on introduction of negatively charged side-chains.

Several of the eIF1A variants were expressed at lower than WT levels, including K7A, K7D, K10D, and ΔG8ΔG9 (*Figure 4A*), as noted above for UM substitutions K4D, T6D, T6R, and ΔG8 (*Figure 2C*). To determine whether the reduced eIF1A expression contributed to their mutant phenotypes, we expressed the subset of variants with the strongest phenotypes from high-copy (hc) plasmids and re-examined their effects on initiation fidelity. The mutant proteins K4D, ΔG8, ΔG8ΔG9 and K10D were expressed from hc plasmids at levels exceeding that of WT eIF1A expressed from a single-copy plasmid (scWT); however, they all still conferred reduced levels of eIF1 expression compared to cells containing normal (scWT) or overexpressed levels of WT eIF1A (hcWT) (*Figure 4—figure supplement 1A*). The overexpressed variants also conferred reduced expression of the *SUI1-lacZ* fusions with native or poor context (*Figure 4—figure supplement 1B*); and they co-suppressed the Sui$^-$/His$^+$ phenotype, elevated UUG:AUG ratio and derepressed *GCN4-lacZ* expression conferred by *SUI3-2* (*Figure 4—figure supplement 2*). We conclude that the reduced expression of eIF1A NTT variants has little impact on their ability to increase discrimination against poor initiation sites in vivo.

To obtain additional support for the conclusion that eIF1A NTT substitutions increase discrimination against AUGs in poor context, we assayed their effects on *GCN4-lacZ* reporters containing a

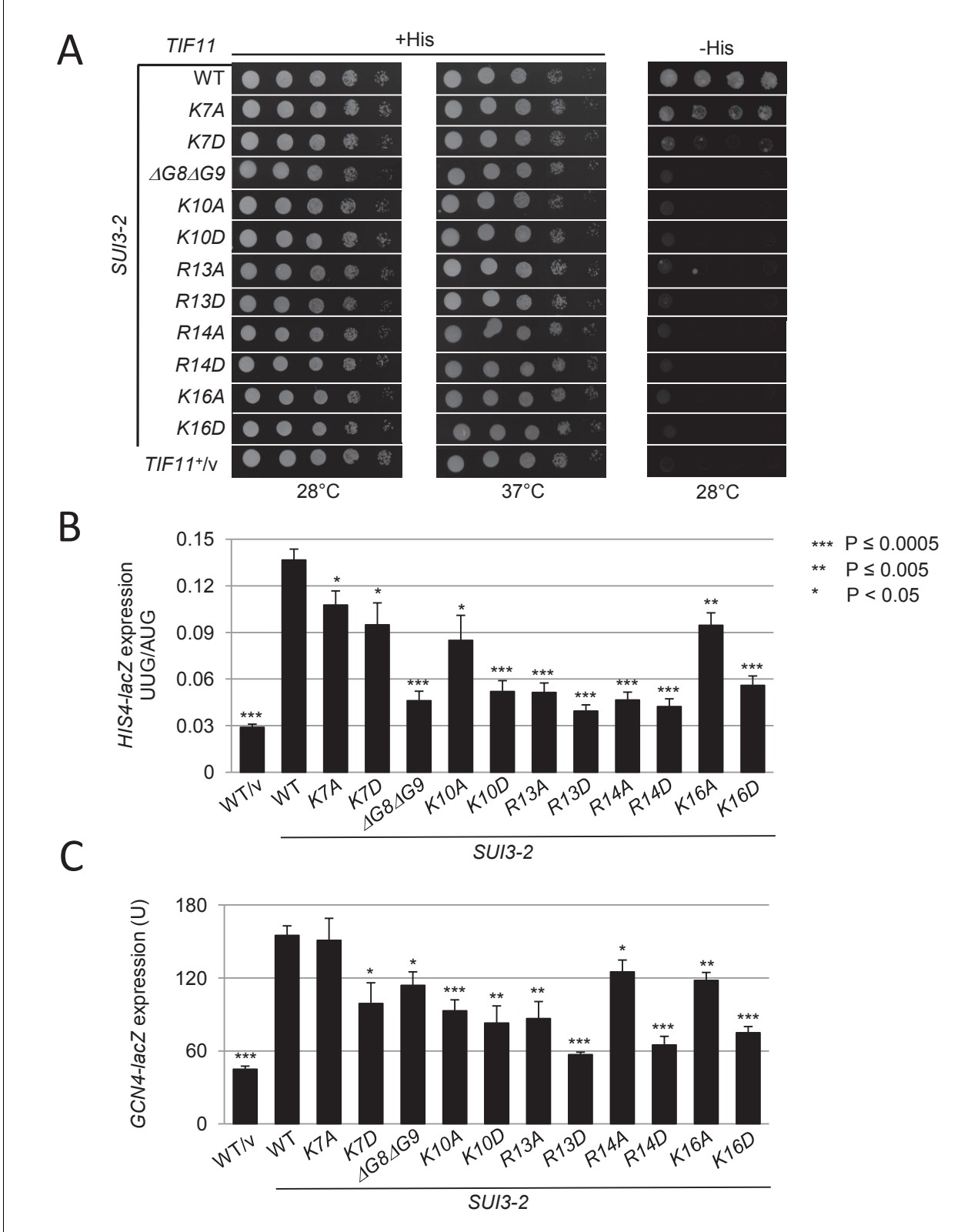

**Figure 3.** Substitutions in key eIF1A NTT basic residues reduce the elevated UUG initiation and derepressed *GCN4-lacZ* expression conferred by Sui⁻ mutation *SUI3-2*. (**A**) Derivatives of strain H3582 containing the indicated *TIF11* alleles and episomal *SUI3-2* (p4280/YCpSUI3-S264Y-W) or empty vector (/v) were analyzed for Slg⁻ and His⁺/Sui⁻ phenotypes by spotting 10-fold serial dilutions on SC-Leu-Trp plus 0.3 mM His and incubated at 28°C or 37°C for 2 days (+His), or on SC-Leu-Trp plus 0.003 mM His (-His) and grown at 28°C for 7 days, as in *Figure 2—figure supplement 1B*. (**B–C**) Transformants

*Figure 3 continued on next page*

*Figure 3 continued*

of the strains from (**A**) harboring *HIS4-lacZ* reporters with AUG or UUG start codons (**B**) or the *GCN4-lacZ* reporter (**C**) were cultured and assayed for β-galactosidase activities as in *Figure 2B*.

DOI: https://doi.org/10.7554/eLife.31250.006

The following source data and figure supplement are available for figure 3:

**Source data 1.** Source data for *Figure 3* and *Figure 3—figure supplement 1*.

DOI: https://doi.org/10.7554/eLife.31250.008

**Figure supplement 1.** Certain eIF1A NTT substitutions affecting PIC interactions confer slow-growth phenotypes andConfirmation of Ssu-phenotypes of R13P and K16D substitutions using UUG- or AUG-initiated luciferase reporters.

DOI: https://doi.org/10.7554/eLife.31250.007

modified upstream ORF1 elongated to overlap the *GCN4* ORF (el.uORF1). In budding yeast, adenines are preferred at positions $-1$ to $-3$ upstream of the AUG (numbered $+1$ to $+3$), with an extremely high occurrence of A and low occurrence of C/U at $-3$; whereas U is preferred at $+4$ (*Shabalina et al., 2004*; *Zur and Tuller, 2013*). With the WT preferred context of $A_{-3}A_{-2}A_{-1}$ present at el.uORF1, virtually all scanning ribosomes recognize this AUG codon (uAUG-1), and because reinitiation at the *GCN4* ORF downstream is nearly non-existent, *GCN4-lacZ* expression is extremely low (*Grant et al., 1994*). Consistent with previous results (*Visweswaraiah et al., 2015*), replacing only the optimal $A_{-3}$ with U increases the bypass (leaky scanning) of uAUG-1 to produce an ~15 fold increase in *GCN4-lacZ* translation, whereas introducing the poor context $U_{-3}U_{-3}U_{-1}$ further increases leaky scanning for a ~50 fold increase in *GCN4-lacZ* expression. Eliminating uAUG-1 altogether increases *GCN4-lacZ* expression by >100 fold (*Figure 4C*, column 1, rows 1–4). From these results, the percentages of scanning ribosomes that either translate el.uORF1 or leaky-scan uAUG-1 and translate *GCN4-lacZ* instead can be calculated (*Figure 4C*, cols. 4 and 7; see legend for details), revealing that >99%, ~89%, and ~66% of scanning ribosomes recognize uAUG-1 in optimum, weak, and poor context, respectively, in WT cells (*Figure 4C* col. 7, rows 1–3).

The UM-associated NTT mutation *R13P* increases leaky scanning of uAUG-1, as indicated by increased *GCN4-lacZ* expression for all three reporters containing el.-uORF1 but not for the uORF-less reporter (*Figure 4C*, cf cols. 1–2, rows 1–4). Calculating the percentages of ribosomes that recognize uAUG-1 revealed that *R13P* (i) conferred the greatest reduction in recognition of uAUG-1 when the latter resides in poor context, from ~66% to ~27%, (ii) produced a moderate reduction for the weak-context reporter, from ~89% to ~77%, and (iii) evoked only a slight reduction when uAUG-1 is in optimal context, from >99% to ~98% (*Figure 4C*, cf. cols. 7–8, rows 1–3). Similar results were obtained for the targeted mutation *R14A* (*Figure 4C*, cf. cols 7 and 9, rows 1–3); and for the targeted *K16A* and *K16D* mutations, with the Asp versus Ala replacement conferring the greater reduction in uAUG-1 recognition (*Figure 4—figure supplement 3A*, cf. cols. 7–9); and also for the hc$\Delta G8\Delta G9$ and hc*K10D* mutations (*Figure 4—figure supplement 3B*, cols. 7–9). Thus, both targeted and UM-associated NTT mutations decrease recognition of AUG start codons by scanning PICs preferentially when they reside in poor versus optimum context.

## eIF1A NTT substitutions R13P and K16D destabilize the closed, $P_{IN}$ conformation of the 48S PIC in vitro

The multiple defects in start codon recognition conferred by the eIF1A NTT mutations suggest that they destabilize the $P_{IN}$ state of the 48S PIC at both UUG and AUG start codons. We tested this hypothesis by analyzing the effects of the R13P and K16D substitutions on the rate of TC dissociation from PICs reconstituted in vitro. Partial 43S·mRNA complexes (lacking eIF3 and eIF5; henceforth p48S PICs) were formed by incubating WT TC (assembled with [$^{35}$S]-Met-tRNA$_i$ and non-hydrolyzable GTP analog GDPNP) with saturating amounts of eIF1, WT or mutant eIF1A, an uncapped unstructured model mRNA containing either AUG or UUG start codon [mRNA(AUG) or mRNA(UUG)], and 40S subunits. p48S PICs containing [$^{35}$S]-Met-tRNA$_i$ were incubated for increasing time periods in the presence of an excess of unlabeled TC (chase) and then resolved via native gel electrophoresis to separate 40S-bound and unbound fractions of TC. From previous work, it was determined that TC bound in the $P_{OUT}$ state is too unstable to remain associated with the PIC during the native gel electrophoresis used to separate PIC-bound from unbound TC in this assay. It was also deduced that a large proportion of WT complexes formed with mRNA(AUG) achieve a highly stable

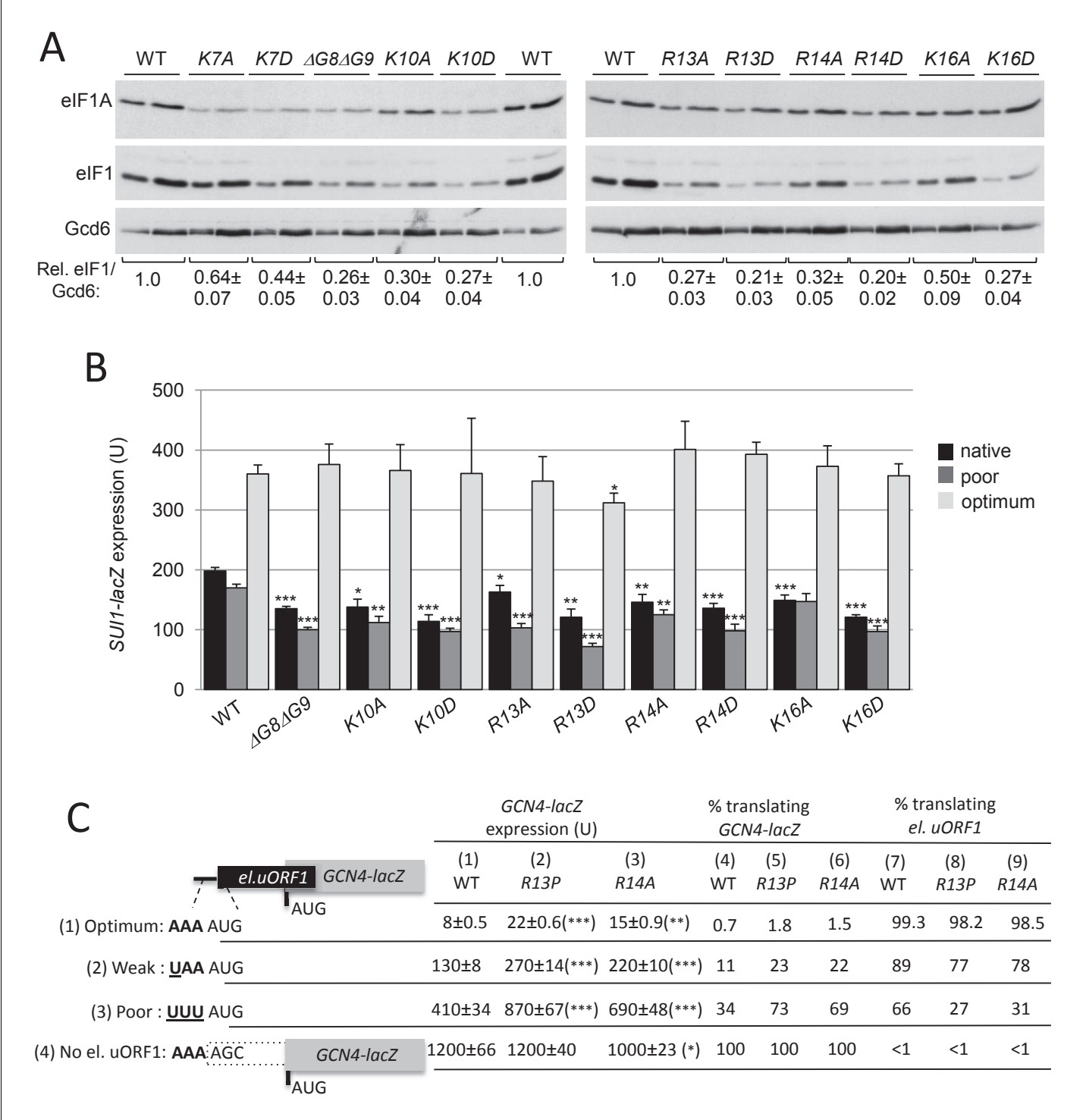

**Figure 4.** Substitutions in key eIF1A NTT basic residues increase discrimination against poor AUG contexts. (**A**) Western blot analysis of eIF1 expression in derivatives of H3582 with the indicated *TIF11* alleles, as in *Figure 2C*. (**B**) Transformants of strains in (**A**) with *SUI1-lacZ* fusions containing the native suboptimal, poor or optimum AUG contexts were assayed for β-galactosidase activities as in *Figure 2D*. (**C**) H3582 derivatives, harboring WT, *R13P* or *R14A TIF11* alleles and *el.uORF1 GCN4-lacZ* reporters containing the depicted optimum (pC3502, row1), weak (pC4466, row2) or poor (pC3503, row3) context of uAUG-1, or uORF-less *GCN4-lacZ* reporter with a mutated uAUG-1 (pC3505, row4), were assayed for β-galactosidase activities as in *Figure 2D*. Mean expression values with S.E.M.s were determined from four transformants (columns 1, 2 and 3). The percentages of scanning ribosomes that translate el.uORF1 (columns 7, 8 and 9) or leaky-scan uAUG-1 and translate *GCN4-lacZ* instead (columns 4, 5 and 6) were

*Figure 4 continued on next page*

*Figure 4 continued*

calculated from results in columns 1, 2 and 3 by comparing the amount of expression observed for each uORF-containing reporter to the uORF-less construct. Statistically significant differences between mutant and WT are marked with asterisks (*p<0.05; **p<0.005; ***p<0.0005).

DOI: https://doi.org/10.7554/eLife.31250.009
The following source data and figure supplements are available for figure 4:

**Source data 1.** Source data for *Figure 4* and *Figure 4—figure supplements 1*, *2* and *3*.
DOI: https://doi.org/10.7554/eLife.31250.013
**Figure supplement 1.** Overexpression of selected eIF1A NTT variants confers reduced expression of eIF1 and *SUI1-lacZ* fusions with native and poor AUG contexts.
DOI: https://doi.org/10.7554/eLife.31250.010
**Figure supplement 2.** Selected eIF1A NTT variants when overexpressed still suppress the His+/Sui- phenotype and elevated UUG initiation and *GCN4-lacZ* expression conferred by *SUI3-2*.
DOI: https://doi.org/10.7554/eLife.31250.011
**Figure supplement 3.** Additional eIF1A targeted and UM-associated NTT mutations increase leaky scanning of *GCN4* uAUG-1 in vivo.
DOI: https://doi.org/10.7554/eLife.31250.012

state from which no TC dissociation occurs during the time-course. A smaller fraction of complexes formed with mRNA (UUG) achieves this highly stable state, and the remainder dissociates with a measurable off-rate (*Kolitz et al., 2009*; *Dong et al., 2014*; *Martin-Marcos et al., 2014*).

In agreement with previous findings, little TC dissociation occurred from the WT PICs formed with mRNA(AUG) over the time course of the experiment (*Figure 5A*), whereas appreciable dissociation was observed from WT PICS assembled with mRNA(UUG) ($k_{off} = 0.18 \pm 0.02$ h$^{-1}$ (*Figure 5A*). Both eIF1A substitutions R13P and K16D increased the extent and rate of TC dissociation from PICs assembled on mRNA(UUG), while having little effect on the mRNA(AUG) complexes (*Figure 5A*). The extent of dissociation reflects the proportion of PICs in $P_{IN}$ versus the hyperstable conformation, and the rate of dissociation reflects the stability of the $P_{IN}$ conformation (*Kolitz et al., 2009*; *Dong et al., 2014*). Thus, our results indicate that the eIF1A substitutions R13P and K16D decrease the fraction of the PICs in the hyper-stable conformation and also destabilize the $P_{IN}$ conformation specifically at near-cognate UUG codons. These findings help to account for the decreased utilization of UUG codons (Ssu- phenotype) conferred by these mutations in vivo.

We also examined the effects of the eIF1A R13P and K16D substitutions on PIC conformation by measuring their effects on the stability of eIF1A binding to the complex. PICs assembled with C-terminally fluorescently-labeled eIF1A were challenged with excess unlabeled eIF1A and the dissociation of labeled eIF1A was monitored over time by following the change in fluorescence anisotropy. The rate of dissociation is generally slow and not physiologically relevant, but reflects the ratio of open to closed PIC conformations and the overall stability of each state (*Maag et al., 2006*; *Fekete et al., 2007*). The dissociation kinetics are usually biphasic, with the fast and slow phases representing dissociation from the open and closed states, respectively; and the magnitude of each rate constant ($k_1$ and $k_2$, respectively) reflects the summation of multiple contacts of eIF1A with the PIC. The ratio of amplitudes of the slow phase to the fast phase ($K_{amp} = \alpha2/\alpha1$) provides an apparent equilibrium constant between the two states, with $K_{amp}$values > 1 indicating a more prevalent closed state. The anisotropy of the labeled eIF1A in the PIC ($R_b$) indicates rotational freedom of the eIF1A CTT, with a higher value indicating greater constraint, which characterizes the closed state.

As expected, WT eIF1A dissociates more slowly from PICs reconstituted with mRNA(AUG) versus mRNA(UUG) (*Figure 5B–C*, WT) with both smaller $k_2$ and larger $K_{amp}$ values, indicating relatively greater occupancy and increased stability of the closed state at AUG. Consistently, the $R_b$ value is greater for mRNA(AUG) versus mRNA(UUG) (*Figure 5D*) (Different batches of fluorescein were employed in labeling matched WT and R13P versus WT and K16D proteins, resulting in different $R_b$ values for the two WT samples). Both the R13P and K16D substitutions dramatically increase the rate of eIF1A dissociation for both mRNAs (*Figure 5B–C*), and evoke monophasic dissociation kinetics with rate constants ($k_1$) much greater than the WT $k_2$ values for both mRNA(AUG) and mRNA(UUG) (*Figure 5B–D*). The $R_b$ values also were reduced by both R13P and K16D using either mRNA. These results indicate that both eIF1A NTT substitutions dramatically destabilize the closed conformation of the PIC at both AUG or UUG start codons.

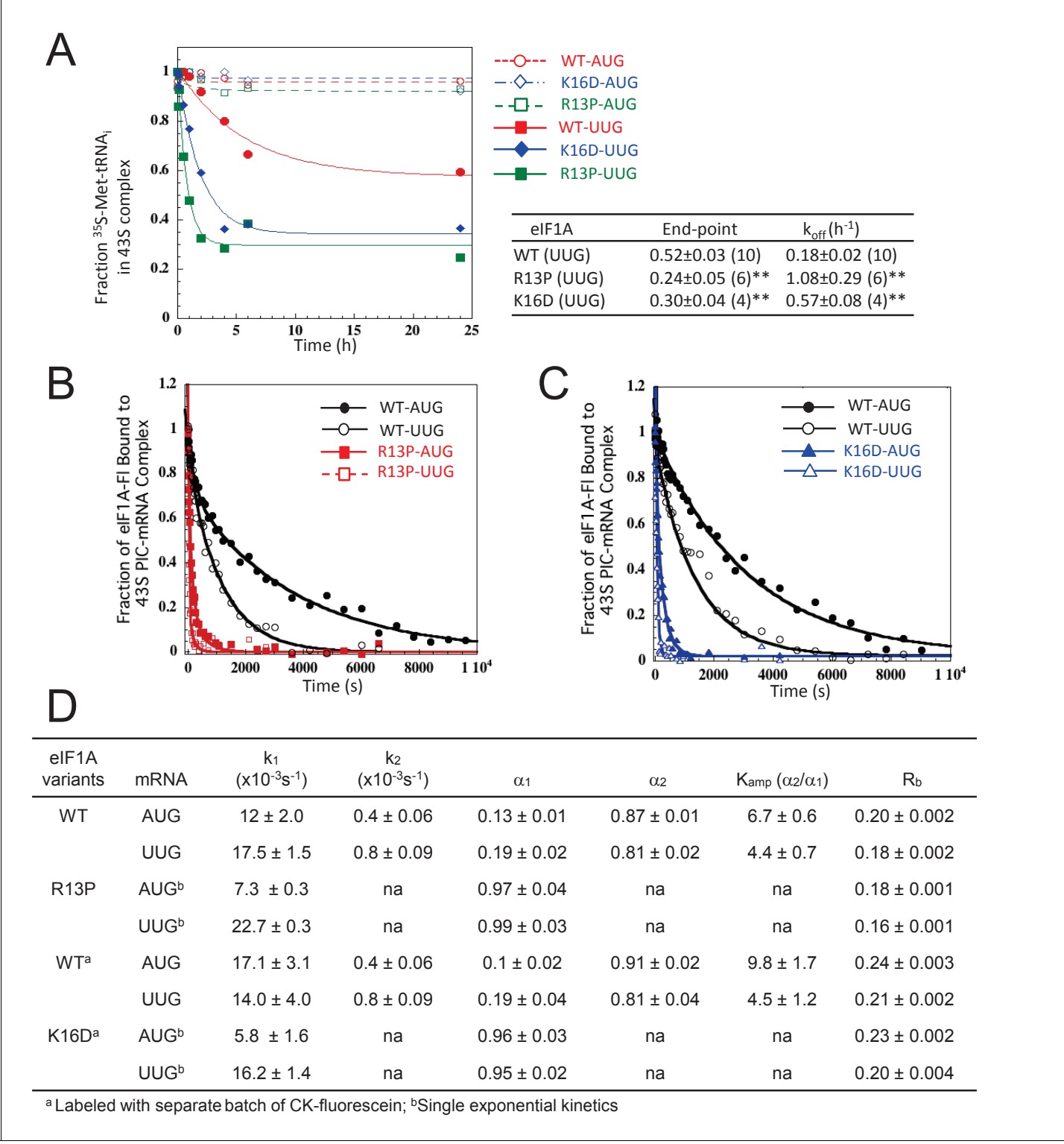

**Figure 5.** UM-associated mutant eIF1A-R13P and targeted mutant eIF1A-K16D destabilize the closed/$P_{IN}$ conformation of the 48S PIC at UUG codons in vitro. (**A**) Effects of R13P and K16D on TC dissociation kinetics from reconstituted partial 43S•mRNA(AUG) or mRNA(UUG) complexes formed with TC containing [$^{35}$S]-Met-tRNA$_i^{Met}$ and WT eIF1A, eIF1A-R13P or eIF1A-K16D, as indicated. Representative curves are shown for each measurement. Tabulated rate constants ($k_{off}$) and reaction end-points with S.E.M.s are averages of between 4–10 replicate experiments (number in parenthesis); asterisks indicate significant differences between mutant and WT as judged by a two-tailed, unpaired Student's t-test (*p<0.05; **p<0.01). (**B–D**) Effects of R13P and K16D on the dissociation kinetics of fluorescein-labeled eIF1A from reconstituted partial 43S•mRNA complexes, monitored by following

*Figure 5 continued on next page*

*Figure 5 continued*

changes in fluorescence anisotropy over time after addition of a large excess of unlabeled WT eIF1A. The data for WT eIF1A were fit with a double exponential decay equation, where the fast phase (rate constant $k_1$) corresponds to dissociation of eIF1A from the 'open' conformation of the PIC and the second phase (rate constant $k_2$) corresponds to dissociation from the 'closed' state (*Maag et al., 2006*). The ratio of amplitudes of the second phase ($\alpha_2$, closed state) to the first phase ($\alpha_1$, open state) is defined as $K_{amp}$. Data for both R13P/K16D were fit to a single exponential equation with rate constant $k_1$. $R_b$ is the anisotropy of PIC-bound eIF1A. (**B–C**) Representative eIF1A dissociation kinetics from PICs assembled with WT (circles), R13P (squares, panel B), or K16D (triangles, panel C) with mRNA(AUG) (filled symbols) or mRNA(UUG) (open symbols). (**D**) Summary of kinetic parameters from experiments in (**B–C**). Different preparations of labeled WT eIF1A were employed for the experiments examining R13P and K16D, as indicated. All experiments were performed at least two times and errors are average deviations.

DOI: https://doi.org/10.7554/eLife.31250.014

The following source data and figure supplement are available for figure 5:

**Source data 1.** Source data for *Figure 5* and *Figure 5—figure supplement 1*.
DOI: https://doi.org/10.7554/eLife.31250.016

**Figure supplement 1.** UM mutant eIF1A-R13P suppresses the stabilizing effect of eIF2 Sui⁻ variant containing eIF2ß-S264Y on the closed conformation of the 48S PIC at UUG codons in vitro.
DOI: https://doi.org/10.7554/eLife.31250.015

Finally, we examined the effects of R13P on eIF1A dissociation kinetics using eIF2 containing the eIF2ß-S264Y variant encoded by *SUI3-2*. In PICs containing mRNA(UUG) and WT eIF1A, eIF2ß-S264Y decreased $k_2$ and increased $K_{amp}$ compared to fully WT PICs, indicating greater occupancy and stability of the closed complex at UUG (*Figure 5—figure supplement 1*, cf. rows 2–3)—which is consistent with the increased UUG initiation (Sui⁻ phenotype) conferred by *SUI3-2* in vivo. Remarkably, both effects of eIF2ß-S264Y on eIF1A dissociation were reversed on replacing WT eIF1A with the R13P variant, and the $R_b$ value was also reduced (*Figure 5—figure supplement 1*, cf. rows 3–4). These findings help to account for the decreased initiation at UUG codons (Ssu⁻ phenotype) conferred by the eIF1A R13P substitution in *SUI3-2* cells (*Figure 2B*). The destabilization of AUG complexes produced by R13P and K16D in the presence of WT eIF2 (*Figure 5B–D*) also helps to explain the increased leaky scanning of AUG codons in poor context evoked by these eIF1A substitutions in otherwise WT cells (*Figures 2C–D* and *4A–C*, and *Figure 4—figure supplement 3A–B*).

## eIF1A-R13P increases discrimination against start codons with non-preferred Kozak context genome-wide

To examine effects of the UM-associated R13P substitution in the yeast translatome, we conducted ribosomal footprint profiling of the *R13P* mutant and isogenic WT strain. This technique entails deep-sequencing of mRNA fragments protected from RNase digestion by translating 80S ribosomes arrested in vitro with cycloheximide (Ribo-seq) in parallel with total mRNA sequencing (RNA-seq) (*Ingolia et al., 2012*). The translational efficiency (TE) of each mRNA is calculated for each strain as the ratio of sequencing reads for ribosome-protected fragments (RPFs) to total mRNA fragments and normalized to the median ratio among all mRNAs, which is assigned a value of unity. Thus, it should be understood that all TE values are expressed relative to the median TE in that strain. Both ribosome footprinting and RNA-seq results were highly reproducible between the two biological replicates for each strain (Pearson's R > 0.99) (*Figure 6—figure supplement 1A–D*). In accordance with the reduced expression of eIF1 conferred by *R13P* (*Figure 2C*, eIF1), both RPF and mRNA reads across the *SUI1* coding sequences (CDS) were diminished in *R13P* cells (*Figure 6A*). Consistent with these results, we showed previously that the reduced translation of *SUI1* mRNA in eIF1 Ssu⁻ mutants evoked by diminished recognition of its poor-context AUG codon is accompanied by reduced *SUI1* mRNA abundance (*Martin-Marcos et al., 2011*). Examples of three other genes with poor context exhibiting reduced translation in *R13P* cells are shown in *Figure 6—figure supplement 2A–C*, which in one case (*SKI2*) also is accompanied by reduced mRNA levels.

To determine whether *R13P* evokes widespread discrimination against AUG codons in poor context, we calculated the changes in TE in mutant versus WT cells as the ratio of $TE_{R13P}$ to $TE_{WT}$ (abbreviated $\Delta TE_{R13P}$) for groups of genes with different Kozak context. Interestingly, *R13P* conferred a moderate, but significant reduction in TE ($\log_2\Delta TE_{R13P}<0$) for a group of 123 genes that contain non-preferred bases at every position surrounding the AUG codon, that is $(C/U/G)_{-3}(C/U/G)_{-2}(C/U/G)_{-1}(aug)(C/A)_{+4}$, (*Shabalina et al., 2004*) (*Zur and Tuller, 2013*) that were selected from a set

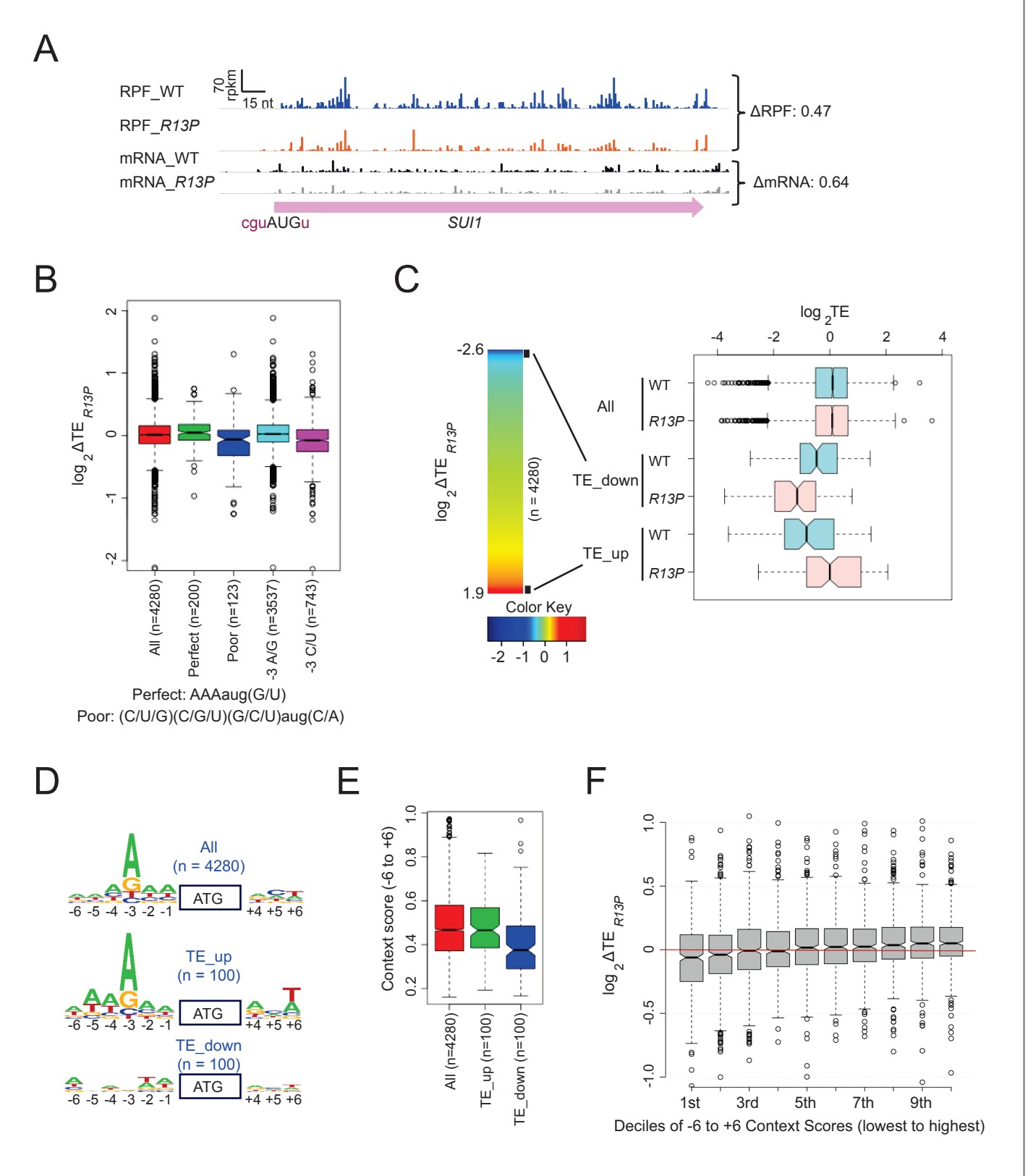

**Figure 6.** UM mutant eIF1A-R13P increases discrimination against poor Kozak context of main CDS AUG codons genome-wide. (**A**) Ribosome-protected fragments (RPFs) and mRNA reads on the *SUI1* gene in WT and *R13P* cells in units of rpkm (reads per 1000 million mapped reads), showing schematically the position of the CDS (pink) and the −3 to −1 and +4 context nucleotides of the AUG codon (in brick). ΔRPF and ΔmRNA give the

*Figure 6 continued on next page*

Figure 6 continued

ratios of RPFs and total mRNA fragments, respectively, in *R13P* versus WT cells for *SUI1*. The Integrated Genomics Viewer (Broad Institute) was employed to display ribosome/mRNA reads. (B) Notched box-plot of the ratios of $\log_2$TE values in *R13P* vs. WT cells ($\Delta TE_{R13P}$) for groups of genes (number, n, indicated) with perfect or poor AUG context (as defined in figure), preferred (A/G) or non-preferred (C/U) bases at −3, and all 4280 genes with >10 RPF reads and >32 mRNA reads (average of 4 samples, two replicates of WT and two replicates of *tif11-R13P*) in the main CDS, and 5′UTR length >5 nt. (C) *left:* Heat-map of TE changes in *R13P* versus WT cells for 4280 genes. Black boxes at the top and bottom of the map demarcate the groups of 100 genes designated as TE_down and TE_up, respectively. *right:* Box-plots of $\log_2$TE values in *R13P* versus WT cells for the 'TE_down' and 'TE_up' groups of genes. (D) Logos of AUG context sequences for the 4280 genes in (B), and the 'TE_up' and 'TE_down' groups of genes defined in (C). (E) Box-plots of AUG context scores calculated for positions −6 to −1 and +4- + 6 for the 'TE_up' and 'TE_down' groups of genes. (F) Box-plot analysis of $\Delta TE_{R13P}$ values for the same 4280 genes analyzed in (B–E) for deciles of equal size binned according to the AUG context scores calculated as in (E).

DOI: https://doi.org/10.7554/eLife.31250.017

The following figure supplements are available for figure 6:

**Figure supplement 1.** Genome-wide ribosome footprint and mRNA reads for WT and *tif11-R13P* strains.

DOI: https://doi.org/10.7554/eLife.31250.018

**Figure supplement 2.** Supporting evidence that eIF1A-R13P increases discrimination against poor Kozak context of main CDS AUG codons genome-wide.

DOI: https://doi.org/10.7554/eLife.31250.019

**Figure supplement 3.** Supporting evidence that eIF1A-R13P increases discrimination against poor Kozak context of both main CDS and uORF AUG codons.

DOI: https://doi.org/10.7554/eLife.31250.020

**Figure supplement 4.** Changes in TE conferred by *R13P* are not correlated with 5′UTR length or propensity for structure.

DOI: https://doi.org/10.7554/eLife.31250.021

of 4280 genes with adequate read-depth and annotated 5′UTR lengths of $\geq$5 nt (*Figure 6B*, 'Poor' context vs 'All'). The same was true for a larger group of 743 genes containing the least preferred bases C/U at the critical −3 position regardless of the sequence at other positions (*Figure 6B*, '−3 C/U' vs 'All'). By contrast, for 200 genes with the preferred bases at all positions, ie. $A_{-3}A_{-2}A_{-1}(AUG)$ $(G/U)_{+4}$, designated 'Perfect' context, or for 3537 genes with A/G at −3, we observed a modest increase in $\Delta TE_{R13P}$ values, compared to all genes (*Figure 6B*, 'Perfect', '−3A/G' vs. 'All'). Knowing that changes in *SUI1* mRNA translation are associated with changes in mRNA abundance in the same direction, we repeated the analysis in *Figure 6B* considering changes in RPFs rather than TE in the mutant cells, and obtained essentially identical results (*Figure 6—figure supplement 2D*). These findings indicate that *R13P* increases discrimination against AUG start codons with non-preferred Kozak context at many genes in the manner observed for the *SUI1* AUG (*Figure 2C–D*), while conferring an increase in TE for mRNAs with preferred context. Examples of genes exhibiting a relative increase in translation in *R13P* cells are presented in *Figure 6—figure supplement 3* (panels A-C).

As an orthogonal approach to detecting increased discrimination against poor context by the *R13P* mutation, we sorted genes on the magnitude of $\Delta TE_{R13P}$ values to identify two subsets of genes exhibiting the greatest increases or decreases in TE in mutant cells. As shown in the heat-map of *Figure 6C*, there are widespread decreases or increases in TE in *R13P* versus WT cells involving thousands of genes, spanning an ~23 fold range of $TE_{WT}/TE_{R13P}$ ratios from 0.16 to 3.73. We focused on the 100 genes showing the greatest decreases or increases in TE in the mutant versus WT (demarcated with boxes at the top and bottom of the heat-map in *Figure 6C*, respectively). The median TE values of these two groups, designated 'TE_down' and 'TE_up', differ significantly between mutant and WT cells (*Figure 6C*, boxplots). Constructing sequence logos for positions −6 to −1 and +4 to +6 for these groups of genes revealed that TE_up genes exhibit sequence preferences highly similar to the consensus sequence observed for all 4280 genes (*Figure 6D*, TE_up vs. All genes), whereas the TE_down genes lack the strong preference for A/G at −3, as well as the moderate preferences for A at −5 and −2 exhibited by the TE_up group of genes (*Figure 6D*, TE_down vs TE_up). We then calculated the AUG context adaptation scores for these sets of genes (*Miyasaka, 1999*), quantifying the similarity between the context of each gene to that of the 2% of all yeast genes with the highest ribosomal load (*Zur and Tuller, 2013*). Context scores among all yeast genes range from ~0.16 (poorest) to ~0.97 (best), with the most highly expressed genes in yeast exhibiting scores near the top of this range. The 100 genes in the TE_down group exhibit context scores significantly below the median score of ~0.47 for all genes, whereas the context scores for

genes in the TE_up group do not differ significantly from the genome-average score (*Figure 6E*). Finally, comparing $\Delta TE_{R13P}$ values for 10 deciles of all 4280 genes divided into bins of equal size according to context scores revealed a continuous decline in $\Delta TE_{R13P}$ progressing from bins with the highest to lowest context scores (*Figure 6F*).

The correlation between the TE changes conferred by *R13P* and AUG context score shown in *Figure 6E–F* was identified without taking into account whether the genes exhibit statistically significant differences in TE between mutant and WT cells. Because too few such mRNAs were identified for rigorous analysis of the correlation, we examined two groups of ~150–200 genes exhibiting significant changes in ribosome occupancy across the CDS between mutant and WT cells (FDR < 0.1). The 159 genes showing a decrease in ribosome occupancy in *R13P* cells exhibit significantly lower context scores, whereas 214 genes with elevated ribosome occupancies display higher context scores, compared to all 4307 genes examined (*Figure 6—figure supplement 2E*).

Together, the results indicate that genes with AUG codons in poor context tend to exhibit reductions in TE in *R13P* cells throughout the yeast translatome. The increases in TE observed for genes with preferred context in the mutant might result from decreased competition for limiting initiation factors or 40S subunits owing to reduced translation of mRNAs with poor context. Alternatively, it might partially reflect the normalization of total RPFs and mRNA reads between mutant and WT cells, which sets the average TE value to unity in each strain, such that decreases in TE for one group of genes is necessarily matched by increases in TE for other genes.

We asked next whether changes in TE (or RPFs) conferred by *R13P* might involve other features of the initiation region, including its propensity for forming secondary structures or proximity of the AUG codon to the 5' end of the mRNA—both parameters associated with reduced initiation efficiency in WT cells (*Kozak, 1991*; *Kertesz et al., 2010*; *Hinnebusch, 2011*; *Ding et al., 2012*). To examine the possible contribution of structure, we interrogated a published database (*Kertesz et al., 2010*) wherein each transcribed nucleotide in 3000 different yeast transcripts was assigned a 'parallel analysis of RNA structure' (PARS) score, based on its susceptibility in mRNA reannealed in vitro to digestion with nucleases specific for single-stranded or double-stranded RNA, with a higher PARS score denoting a higher probability of double-stranded conformation. For each transcript, we tabulated the average PARS score over the entire 5'UTR (Average PARS), the sum of PARS scores for the 30nt surrounding the start codon (for genes with a 5' UTR of $\geq$16 nt (dubbed 'Start30 PARS'), and the sum of PARS scores for the 30nt centered on the +15 (Plus15) or +30 nucleotides (Plus30) downstream of the AUG. A heat-map depiction of these PARS scores, as well as 5'UTR length, in relation to $\Delta TE_{R13P}$ changes for all 2355 genes with sufficient read-density tabulated in the PARS database revealed no obvious correlation between the magnitude of TE changes conferred by *R13P* and either 5'UTR length or PARS scores (*Figure 6—figure supplement 4A*). Supporting this, we found no significant difference in the $\Delta TE_{R13P}$ values for a group of 90 mRNAs containing 5' UTR lengths less than $\leq$18 nt versus all 5136 genes with annotated 5'UTR lengths (*Figure 6—figure supplement 4B*); and no difference in $\Delta TE_{R13P}$ values between the 1st and 10th deciles of genes binned according to the Start30 or Plus15 PARS scores, representing the two extremes of these PARS scores among all genes (*Figure 6—figure supplement 4C–D*). These results contrast with our previous findings that genes exhibiting reduced TE on inactivation of RNA helicase Ded1 tend to have unusually long and structured 5'UTRs with greater than average PARS scores (*Sen et al., 2015*).

We showed above that the *R13P* mutation decreases translation of the elongated version of *GCN4* uORF1 specifically when the uORF1 AUG codon resides in poor context, increasing translation of the downstream CDS of the *GCN4-lacZ* reporter. Hence, we examined our ribosome profiling data for evidence of widespread changes in translation of AUG-initiated uORFs that is dictated by the sequence context of the uORF start codon. It is known that translation of *CPA1* mRNA, encoding an enzyme of arginine biosynthesis, is repressed by its uORF in arginine-replete cells owing to increased pausing during termination at the uORF stop codon, which attenuates progression of scanning PICs to the main *CPA1* AUG codon and elicits nonsense-mediated degradation (NMD) of the transcript (*Werner et al., 1987*; *Gaba et al., 2005*). The AUG codon of the *CPA1* uORF exhibits a suboptimal context at the −3 and −1 positions, $U_{-3}A_{-2}U_{-1}(aug)U_{+4}$, which is thought to ensure that a fraction of scanning PICs can bypass the uORF and translate *CPA1* at low arginine levels (*Werner et al., 1987*)(*Gaba et al., 2005*). Interestingly, *R13P* increases ribosome occupancy in the CDS by ~60%, while decreasing ribosome occupancy in the uORF by ~10%, for a change in uORF

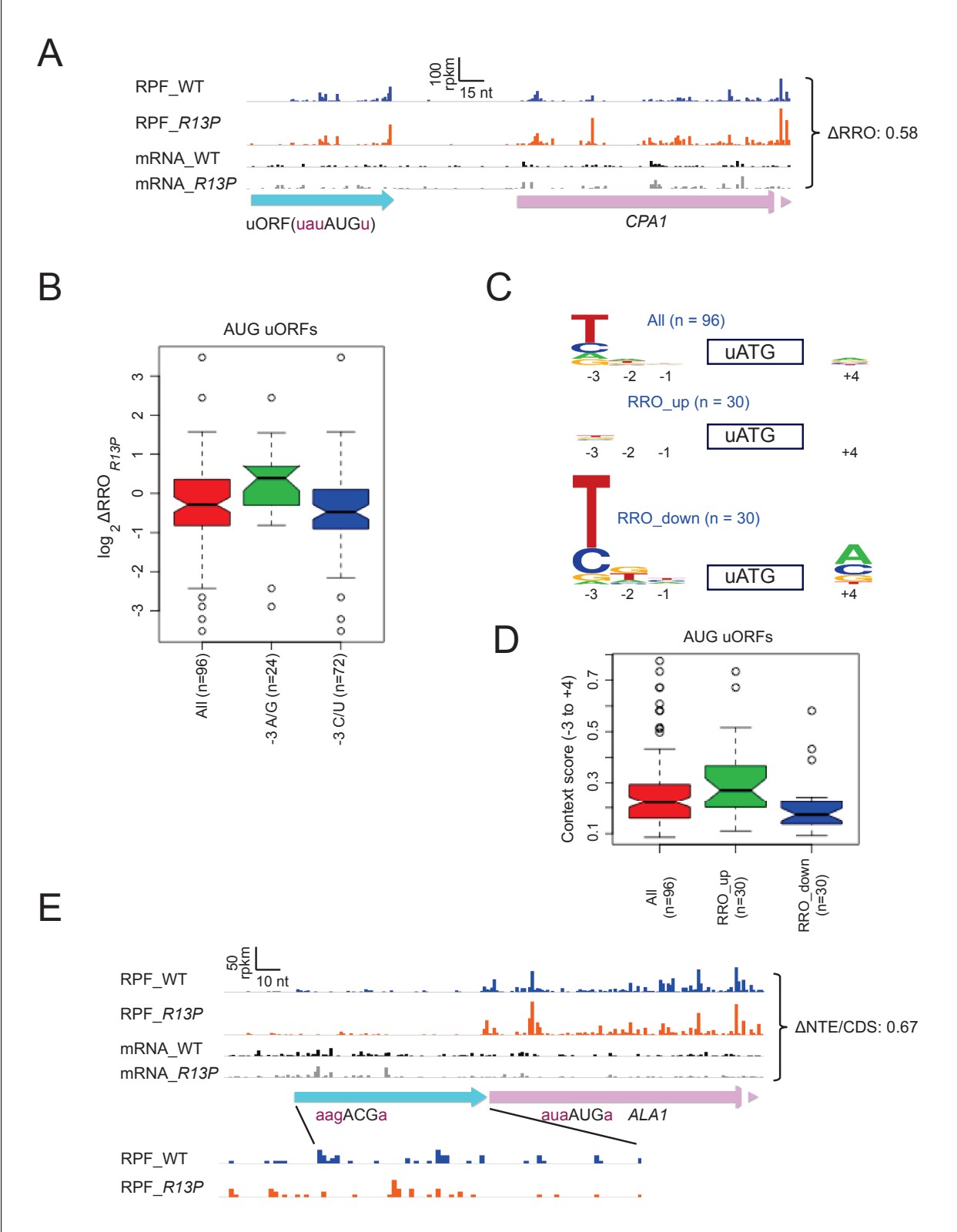

**Figure 7.** eIF1A-R13P increases discrimination against poor Kozak context of uORF AUG codons genome-wide. (**A**) RPFs and mRNA reads on the *CPA1* gene and its uORF with AUG in poor context, displaying a decreased ratio of RPFs in the uORF vs. CDS (RRO) in *R13P* vs. WT cells (ΔRRO = 0.58). The pink arrow missing a portion of the arrowhead designates the beginning of the *CPA1* main CDS. (**B**) Notched box-plot of the ratios of $\log_2$TE values in *R13P* vs. WT cells ($\Delta$TE$_{R13P}$) for a group of 96 genes containing an AUG-initiated uORF and exhibiting >32 RPFs in the main CDS and >2 RPFs

*Figure 7 continued on next page*

*Figure 7 continued*

in the uORF and a 5'UTR for the uORF of >2 nt in length; and of the subsets of 24 genes from this group with preferred A/G at −3, or the 72 genes with non-preferred C/U at −3, relative to the uORF AUG codon. (**C**) Logos of upstream AUG context sequences for the 96 genes in (**B**), and the subsets of 30 genes with the greatest increase (RRO_up) or decrease (RRO_down) in uORF relative to CDS RPFs (RRO values) in *R13P* versus WT cells. (**D**) Box-plots of upstream AUG context scores calculated for positions −3 to −1 and +4 for the same genes analyzed in (**C**). (**E**) RPFs and mRNA reads on the beginning of the *ALA1* main CDS (pink) and N-terminal extension (NTE, cyan schematic), displaying a decreased ratio of NTE/CDS RPFs in *R13P* vs. WT cells (ΔNTE/CDS = 0.67). Note that the ΔNTE/CDS ratio reflects the ratio of initiation at the upstream AUG to the combined initiation events at upstream AUG and main CDS AUG.

DOI: https://doi.org/10.7554/eLife.31250.022

relative to CDS ribosome occupancy (designated as relative ribosome occupancy, 'RRO') of 0.58 (*Figure 7A*), which suggests diminished recognition of the poor-context uORF AUG and attendant increase in CDS translation. An even greater redistribution of ribosomes from uORF to downstream CDS is illustrated for *ICY1* and *BZZ1*, whose uORF AUG codons depart from optimal context at three or all four positions (*Figure 6—figure supplement 3D–E*).

Using bioinformatics, we identified 96 uORFs with AUG start codons that showed evidence of translation in one or more ribosome profiling datasets from WT or various mutant strains, which were obtained in our own laboratory or published by others (see Methods), and which displayed sufficient ribosome occupancies in both the WT and *R13P* strains analyzed here for quantitative analysis. Interestingly, the 72 genes containing uORFs in this group that harbor non-preferred C or U bases at the −3 position mimicked *CPA1* and *ICY1* in showing decreased RRO values in *R13P* versus WT cells, compared to the 24 genes with uORFs containing the preferred bases A or G at −3 (*Figure 7B*). Determination of AUG context logos revealed that the base frequencies for the entire group of 96 uORFs differ markedly from that of AUG codons for main CDSs, exhibiting a preference for U/C versus A/G at −3 and little or no preference at the other positions surrounding the uORF ATG (cf. 'All' in *Figure 7C* vs 'All' in *Figure 6D*), which presumably reflects a need for leaky-scanning of the uORFs. Interestingly, the preference for non-optimal U/C at −3 is even greater, and A is the least prevalent base at −3 for the group of 30 uORFs showing the greatest reductions in RRO in *R13P* cells (*Figure 7C*, RRO_down), which is consistent with increased discrimination against uORF AUGs in poor context in the mutant. By contrast, the preference for non-optimal U/C at −3 is eliminated for the 30 uORFs that exhibit the greatest increases in RRO in *R13P* cells (RRO_up), indicating higher frequencies of the preferred A/G bases at this position for this group of uORFs, which is consistent with decreased discrimination in the mutant against uORF AUGs containing relatively stronger sequence contexts (*Figure 7C*, RRO_up).

Finally, examination of the AUG context scores for all 96 AUG uORFs reveals a much smaller median score (~0.22) (*Figure 7D*) compared to AUGs for all main CDS (~0.47; *Figure 6E*), supporting our conclusion that AUG uORFs as a group exhibit poor context in order to enable leaky scanning in WT cells. Comparing the context scores between two groups of 30 genes exhibiting the greatest increase in RRO (RRO_up) versus the largest decrease in RRO (RRO_down) in the *R13P* versus WT cells supports the tendency for reduced uORF translation in the mutant when the uORF AUG codon is in poor context but increased uORF translation when the uORF AUG has favorable context (*Figure 7D*). Thus, discrimination against suboptimal context contributes to reduced uORF translation, as well as reduced main CDS translation, in *R13P* cells.

The *R13P* mutation increases discrimination against UUG codons in *SUI3-2* and *SUI5* cells (*Figure 2A–B*). We found that in cells lacking a Sui⁻ mutation, *R13P* reduced the *HIS4-lacZ* UUG:AUG initiation ratio by a factor of ~2 (from 0.021 ± 0.002 to 0.011 ± 0.001), smaller than the ~3 fold decrease observed in cells containing *SUI3-2* (*Figure 2B*). Similarly, we found evidence that *R13P* decreases utilization of the near-cognate ACG start codon that initiates the longer, mitochondrial isoform of alanyl-tRNA synthetase encoded by *ALA1*, reducing the ratio of ribosome occupancies in the N-terminal extension relative to the CDS (ΔNTD/CDS) in the mutant to 0.67 of the WT value (*Figure 7E*). This finding is consistent with relaxed discrimination against this native, near-cognate start codon in *R13P* cells.

## Discussion

In this report we show that all seven substitutions in the NTT of yeast eIF1A associated with uveal melanoma in humans confer hyperaccuracy phenotypes in yeast cells. They suppress inappropriate initiation at a UUG start codon in *his4-301* mRNA to prevent growth in the absence of a histidine supplement. They also reduce the UUG: AUG initiation ratio of a *HIS4-lacZ* reporter, in cells harboring the Sui⁻ mutation in eIF2ß (*SUI3-2*) that reduces accuracy and elevates UUG initiation. Like previously identified Ssu⁻ substitutions in eIF1 (*Martin-Marcos et al., 2011*), these eIF1A NTT substitutions also suppress the toxicity of *SUI5* to cell growth at elevated temperatures. Moreover, they decrease initiation at the AUG codons of both *SUI1* mRNA (encoding eIF1) and the *GCN4* uORF1 specifically when they reside in unfavorable Kozak context. The recent structure of a yeast partial 48S PIC predicts that the UM-associated substitutions in the C-terminal portion of the NTT affect direct contacts of the NTT with mRNA nucleotides adjacent to the AUG codon, or in the anticodon of tRNA$_i$, and both interactions are thought to stabilize the PIC in the closed conformation with Met-tRNA$_i$ accommodated in the P$_{IN}$ state (*Hussain et al., 2014*; *Llácer et al., 2015*). Accordingly, the effects of the UM substitutions in reducing near-cognate UUG and poor-context AUG utilization can be attributed to destabilization of the P$_{IN}$ state with attendant increased requirement for a perfect codon-anticodon duplex and optimal context. Two main lines of evidence support this interpretation. First, an identical set of phenotypes was observed for directed substitutions of conserved basic residues in the NTT that also make direct contacts with mRNA or anticodon nucleotides, namely K7, K10, K16, and R14. Substitutions of these residues with Asp have stronger phenotypes than Ala substitutions, consistent with replacing electrostatic attraction (Lys/Arg) with repulsion (Asp) for the phosphodiester backbone of mRNA or tRNA$_i$. The same was true for Asp and Ala substitutions of R13, which is replaced with Pro or His by UM-associated mutations. Second, biochemical experiments reveal that the R13P UM substitution and the directed K16D substitution specifically destabilize the P$_{IN}$ state at UUG codons in vitro, increasing both the fraction of reconstituted PICs from which TC dissociates and the rate of this reaction (k$_{off}$) with a UUG, but not AUG, start codon in the mRNA. These substitutions also increase the rate of eIF1A dissociation, signifying a reduced fraction of PICs in the closed conformation and decreased overall stability of these complexes, with either UUG or AUG start codons. Together, these results help to account for the decreased usage of UUGs and AUGs in poor context conferred by these mutations in vivo, and support the notion that their hyperaccuracy phenotypes result from reduced occupancy and stability of the closed/P$_{IN}$ state that, in turn, confers a heightened requirement for optimal initiation sites.

Although reduced initiation at near-cognate UUG codons in Sui⁻ mutants (Ssu⁻ phenotype) was reported previously for clustered alanine substitutions of eIF1A NTT residues 7–11, 12–16, and 17–21 (*Fekete et al., 2007*), belonging to the scanning inhibitor element designated SI$_1$ (*Saini et al., 2010*), it was unknown which residues in these three segments are most critical for increasing UUG initiation, nor whether any residues in the 7–11 and 12–16 intervals increase initiation at AUGs in poor context. Our findings establish that all five basic residues conserved between yeast and humans that contact mRNA, the anticodon, or 18S rRNA in the decoding center of the py48S PIC (K7, K10, R13, R14, and K16) are critical for efficient utilization of these suboptimal initiation sites, as is the conserved Gly8-Gly9 turn required for making these key contacts (*Figure 1B–C*) (*Hussain et al., 2014*). Accordingly, the Ala substitutions of K7 and K10 generated by the *7–11* mutation, and of R13, R14, and K16 by mutation *12–16* are likely responsible for the Ssu⁻ phenotypes of these multiple-residue substitutions (*Fekete et al., 2007*). Although the *17–21* mutation does not substitute any of the key basic residues, it might impair interactions of the C-terminal section of the eIF1A NTT with PIC components and indirectly prevent the basic residues in the N-terminal portion of the NTT from engaging with the decoding center (*Figure 1B*). Finally, our results implicate eIF1A residues K3 and K4 (N4 in humans), also substituted in UM, in controlling initiation accuracy, but their molecular functions remain to be determined, as they were not resolved in the py48S structure.

In addition to suppressing the elevated UUG initiation (Sui⁻ phenotype) conferred by the eIF2ß mutation *SUI3-2*, the eIF1A NTT substitutions we analyzed also suppress the derepressed *GCN4-lacZ* expression (Gcd⁻ phenotype) produced by *SUI3-2*. eIF1 stabilizes the open conformation of the PIC, to which TC binds most rapidly (P$_{OUT}$ state) (*Figure 1A*) (*Passmore et al., 2007*). The Gcd⁻ phenotypes conferred by other Sui⁻ mutations affecting eIF1, eIF1A, and tRNA$_i$ have been attributed to destabilization of this P$_{OUT}$ state of TC binding. This interpretation was based partly on the finding

that they are suppressed by Ssu⁻ substitutions in the $SI_1$ and $SI_2$ elements of eIF1A that destabilize the closed/$P_{IN}$ conformation and thus shift the system in the opposite direction towards the open/$P_{OUT}$ state, which should accelerate TC binding (*Saini et al., 2010*) (*Dong et al., 2014*; *Martin-Marcos et al., 2014*). Destabilization of the open/$P_{OUT}$ state probably also contributes to the Gcd⁻ phenotype of *SUI3-2* because it is suppressed by Ssu⁻ substitutions in $SI_1$ and $SI_2$ of eIF1A (*Saini et al., 2010*). Thus, the marked co-suppression of the Sui⁻ and Gcd⁻ phenotypes of *SUI3-2* observed here for substitutions of the key basic residues K7, K10, R13, and K16 of the NTT, particularly for the acidic Asp replacements, provides additional genetic evidence that they preferentially destabilize the closed/$P_{IN}$ state and shift the system towards the open conformation to which TC loads during assembly of scanning PICs.

We used ribosome footprint profiling to demonstrate that the R13P UM substitution confers a broad decrease in utilization of AUG codons with poor Kozak context in the yeast translatome, mimicking the effect of R13P in reducing eIF1 synthesis from *SUI1* mRNA. R13P also reduced recognition of a subset of the ~100 uORFs whose translation we detected in both mutant and WT cells when their AUG codons reside in poor context, mimicking the effect of R13P of increasing leaky scanning through the elongated version of *GCN4* uORF1 specifically when its AUG codon resides in poor context. *R13P* cells also display somewhat increased discrimination against the near-cognate ACG start codon of the *ALA1* mRNA that initiates an N-terminal extension of the encoded alanyl tRNA synthetase, decreasing the ratio of reads in the extension versus the CDS by ~1/3rd. *ALA1* is one of only two annotated genes with non-AUG start codons in yeast (*Chen et al., 2008*; *Chang et al., 2010*), and the other such gene, *GRS1,* showed no reduction in initiation from the UUG codon initiating the N-terminal extension of glycyl-tRNA synthetase. This different behavior might be explained by the fact that the context of the *GRS1* UUG matches closely the optimum consensus sequence in yeast in containing A's at −4 to −1, and U at +4, whereas the *ALA1* ACG deviates from this consensus by lacking A's at −1 and −4 and containing A at +4. On the other hand, *R13P* modestly decreased initiation at the UUG codon of the *HIS4-lacZ* reporter, even though it contains preferred A's at −4,–3, and −1. Thus, it remains to be seen whether poor context will be a significant determinant of increased usage of near-cognate start codons conferred by eIF1A NTT Ssu⁻ substitutions.

Considering that the sequence of the yeast and human eIF1A-NTT are quite similar, and that R13 is conserved between the two species (*Figure 1C*), our findings for the UM substitutions in yeast eIF1A lead us to propose that the corresponding substitutions in the human eIF1A NTT will favor oncogenic transformation by increasing discrimination against AUG codons with poor context or near-cognate start codons. If one or more tumor suppressor genes contains such a poor initiation site, the UM substitutions can be expected to increase its relative translation rate and thereby impair one or more control mechanisms governing cell proliferation. A recent study on UM tumor cell lines expressing either WT or the G6D variant of EIF1AX provided evidence that the G6D substitution reduces the critical function of EIF1AX in bulk translation initiation. Interestingly, sequencing of total polysomal mRNA indicated that ribosomal protein genes (RPGs) as a group have a heightened requirement for EIF1AX and that the translation of these mRNAs is reduced in G6D vs WT cells (*Johnson et al., 2017*). Given their high rates of translation during rapid cell growth, it seems likely that RPGs would exhibit favorable Kozak context, and by analogy with our findings in yeast on the eIF1A R13P substitution, the RPGs would not be expected to show decreased translation as the result of heightened discrimination against poor context. However, the yeast equivalent of G6D, T6D, did not significantly increase discrimination against the suboptimal eIF1 AUG codon in yeast in the manner observed for R13P. Moreover, unlike G6D in the tumor cells, we found no evidence that the UM-related substitutions in yeast eIF1A reduce bulk initiation. Thus, it is possible that the reduction in RPG expression in G6D tumor cells is a response to reduced bulk translation and cell growth; and it will be interesting to determine whether the R13P substitution in EIF1AX increases discrimination against AUGs in poor context in human cells.

## Materials and methods

### Plasmid constructions

Plasmids employed in this work are listed in *Table 1*. *TIF11* mutations in plasmids p5633, p5635, p5637, p5638, p5640, p5642 and p5644 were introduced in plasmid p3990 using GeneArtSite-

**Table 1.** Plasmids used in this study

| Plasmid | Description | Source or reference |
|---|---|---|
| YCplac111 | sc *LEU2* cloning vector | (*Gietz and Sugino, 1988*) |
| YEplac181 | hc *LEU2* cloning vector | (*Gietz and Sugino, 1988*) |
| YCplac22 | sc *TRP1* cloning vector | (*Gietz and Sugino, 1988*) |
| p3390/pDSO9 | sc *LEU2 TIF11* in YCplac111 | (*Choi et al., 2000*) |
| p5633 | sc *LEU2 tif11-K3E* in YCplac111 | This study |
| p5635 | sc *LEU2 tif11-K4D* in YCplac111 | This study |
| p5638 | sc *LEU2 tif11-T6D* in YCplac111 | This study |
| p5637 | sc *LEU2 tif11-T6R* in YCplac111 | This study |
| p5640 | sc *LEU2 tif11-ΔG8* in YCplac111 | This study |
| p5642 | sc *LEU2 tif11-R13P* in YCplac111 | This study |
| p5644 | sc *LEU2 tif11-G15D* in YCplac111 | This study |
| pDH469 | sc *LEU2 tif11-K7A* in YCplac111 | This study |
| pDH468 | sc *LEU2 tif11- K7D* in YCplac111 | This study |
| pDH481 | sc *LEU2 tif11-ΔG8ΔG9* in YCplac111 | This study |
| pDH471 | sc *LEU2 tif11-K10A* in YCplac111 | This study |
| pDH470 | sc *LEU2 tif11-K10D* in YCplac111 | This study |
| pDH473 | sc *LEU2 tif11-R13A* in YCplac111 | This study |
| pDH472 | sc *LEU2 tif11-R13D* in YCplac111 | This study |
| pDH475 | sc *LEU2 tif11-R14A* in YCplac111 | This study |
| pDH474 | sc *LEU2 tif11-R14D* in YCplac111 | This study |
| pDH478 | sc *LEU2 tif11-K16A* in YCplac111 | This study |
| pDH476 | sc *LEU2 tif11-K16D* in YCplac111 | This study |
| p3400/pDSO23 | hc *LEU2 TIF11* in YEplac181 | (*Choi et al., 2000*) |
| pPMB167 | hc *LEU2 tif11-K4D* in YEplac181 | This study |
| pPMB168 | hc *LEU2 tif11-ΔG8* in YEplac181 | This study |
| pPMB169 | hc *LEU2 tif11-ΔG8ΔG9* in YEplac181 | This study |
| pPMB170 | hc *LEU2 tif11- K10D* in YEplac181 | This study |
| p4281/YCpTIF5-G31R-W | sc *TRP1 TIF5-G31R* in YCplac22 | (*Valásek et al., 2004*) |
| p4280/YCpSUI3-S264Y-W | sc *TRP1 SUI3-S264Y* in YCplac22 | (*Valásek et al., 2004*) |
| p367 | sc *URA3 HIS4(ATG)-lacZ* | (*Donahue and Cigan, 1988*) |
| p391 | sc *URA3 HIS4(TTG)-lacZ* | (*Donahue and Cigan, 1988*) |
| p180 | sc *URA3 GCN4-lacZ* | (*Hinnebusch, 1985*) |
| pPMB24 | sc *URA3 SUI1-lacZ* | (*Martin-Marcos et al., 2011*) |
| pPMB25 | sc *URA3 SUI1-opt-lacZ* | (*Martin-Marcos et al., 2011*) |
| pPMB28 | sc *URA3 SUI1$_{UUU}$-lacZ* | (*Martin-Marcos et al., 2011*) |
| pC3502 | sc *URA3* $^{-3}$AAA$^{-1}$ el.uORF1 *GCN4-lacZ* in YCp50 | (*Visweswaraiah et al., 2015*) |
| pC4466 | sc *URA3* $^{-3}$UAA$^{-1}$ el.uORF1 *GCN4-lacZ* in YCp50 | (*Visweswaraiah et al., 2015*) |
| pC3503 | sc *URA3* $^{-3}$UUU$^{-1}$ el.uORF1 *GCN4-lacZ* in YCp50 | (*Visweswaraiah et al., 2015*) |
| pC3505 | sc *URA3* el.uORF1-less *GCN4-lacZ* in YCp50 | (*Visweswaraiah et al., 2015*) |
| pTYB2-TIF11 | *TIF11* in pTYB2 | (*Acker et al., 2007*) |
| p6013 | *tif11-R13P* in pTYB2 | This study |
| p6015 | *tif11-K16D in pTYB2* | This study |
| pRaugFFuug | Dual luciferase reporter *LUC$_{ren}$(aug)-LUC$_{firefly}$ (uug)* in *URA3* vector | (*Kolitz et al., 2009*) |

*Table 1 continued on next page*

Table 1 continued

| Plasmid | Description | Source or reference |
|---|---|---|
| pRaugFFuug | Dual luciferase reporter $LUC_{ren}(aug)$-$LUC_{firefly}$ (uug) in *URA3* vector | (*Kolitz et al., 2009*) |

DOI: https://doi.org/10.7554/eLife.31250.023

Directed Mutagenesis System (Invitrogen, ThermoFisher) and the appropriate set of complementary mutagenic oligonucleotide primers listed in Table S1, **Supplementary file 1**, following the manufacturer's instructions except for the use of Phusion High fidelity Polymerase (New England BioLabs) for the first step of amplification. Plasmids pDH468, pDH469, pDH481, pDH471, pDH470, pDH473, pDH472, pDH475, pDH474, pDH478, and pDH476 were derived from p3390 by site-directed mutagenesis using the QuickChange XL kit (Agilent Technologies) and the appropriate primers in Table S1. Plasmids pPMB167 to pPMB170 were created by inserting a ~1.2 kb EcoRI-SalI fragment containing *tif11-K4D*, *tif11-ΔG8*, *tif11-ΔG8ΔG9* and *tif11-K10D* alleles from p5635, p5640, pDH481 and pDH470, respectively, into the corresponding sites of YCplac181. Plasmids p6013 (*tif11-R13P*) and p6015 (*tif11-K16D*) for expression of eIF1A variants for biochemical analyses were made by PCR amplification of the appropriate DNA fragments from plasmids p5642 and pDH476, respectively and insertion of the resulting fragments into the NdeI-XmaI sites of pTYB2. All plasmids were sequenced to verify the presence of the intended mutations.

## Yeast strain constructions

Yeast strains employed in this work are listed in *Table 2*. Derivatives of strain H3582 [*MATa ura3-52 trp1Δ63 leu2-3, leu2-112 his4-301(ACG) tif11Δ* p3392 (sc *URA3 TIF11*)] were constructed by transforming H3582 to Leu$^+$ with single copy (sc) or high copy (hc) *LEU2* plasmids harboring the appropriate *TIF11* alleles on synthetic complete medium (SC) lacking leucine (SC-Leu), and the resident *TIF11$^+$URA3* plasmid (p3392) was evicted by selecting for growth on 5-FOA medium. Derivatives of strain H3582 containing plasmid-borne *SUI5* (p4281/YCpTIF5-G31R-W), *SUI3-2* (p4280/YCpSUI3-S264Y-W) or empty vector were generated by transformation and selection on SC lacking leucine and tryptophan (SC-Leu-Trp). Strains FZY010/FZY011 and PMY337/PMY338 used for ribosome profiling are independent transformants of strains PMY290 and PMY318 with *TRP1* vector YCplac22, respectively.

## Biochemical assays using yeast cell extracts

Assays of β-galactosidase activity in whole cell extracts (WCEs) were performed as described previously (*Moehle and Hinnebusch, 1991*). At least four biological replicates (independent transformants) were employed for all β-galactosidase activity measurements. Unpaired t-tests were performed to compare wild type and mutant mean values and the change was considered significant if the two-tailed P value was < 0.05. Luminescence expressed from dual luciferase reporter plasmids pRaugFFuug and pRaugFFaug was measured as described previously (*Kolitz et al., 2009*). For Western analysis, WCEs from three biological replicates (independent transformants) were prepared by trichloroacetic acid extraction as previously described (*Reid and Schatz, 1982*) and immunoblot analysis was conducted as previously described (*Nanda et al., 2009*) using antibodies against eIF1A/Tif11 (*Olsen et al., 2003*), eIF1/Sui1 (*Valásek et al., 2004*) and eIF2Bε/Gcd6 (*Bushman et al., 1993*). Two technical replicates were performed using the same extracts and two different amounts of each extract differing by 2-fold were loaded in successive lanes. Enhanced chemiluminiscence (Amersham) was used to visualize immune complexes, and signal intensities were quantified by densitometry using NIH ImageJ software.

## Biochemical analysis in the reconstituted yeast system

WT eIF1 and eIF1A and eIF1A variants R13P and K16D were expressed in BL21(DE3) Codon Plus cells (Agilent Technologies) and purified using the IMPACT system (New England Biolabs) as described previously (*Acker et al., 2007*). His$_6$-tagged WT eIF2, or the variant containing eIF2β-S264Y, were overexpressed in yeast strains GP3511 and H4560, respectively, and purified as

**Table 2.** Yeast strains used in this study

| Strain | Genotype | Source |
|---|---|---|
| H3582 | *MATa ura3-52 trp1Δ63 leu2-3, leu2-112 his4-301(ACG) tif11Δ p3392 (sc URA3 TIF11)* | (*Fekete et al., 2005*) |
| PMY318 | *MATa ura3-52 trp1Δ63 leu2-3, leu2-112 his4-301(ACG) tif11Δ p3390 (sc LEU2 TIF11)* | This study |
| PMY284 | *MATa ura3-52 trp1Δ63 leu2-3, leu2-112 his4-301(ACG) tif11Δ p5633 (sc LEU2 tif11-K3E)* | This study |
| PMY285 | *MATa ura3-52 trp1Δ63 leu2-3, leu2-112 his4-301(ACG) tif11Δ p5635 (sc LEU2 tif11-K4D)* | This study |
| PMY286 | *MATa ura3-52 trp1Δ63 leu2-3, leu2-112 his4-301(ACG) tif11Δ p5638 (sc LEU2 tif11-T6D)* | This study |
| PMY287 | *MATa ura3-52 trp1Δ63 leu2-3, leu2-112 his4-301(ACG) tif11Δ p5637 (sc LEU2 tif11-T6R)* | This study |
| PMY289 | *MATa ura3-52 trp1Δ63 leu2-3, leu2-112 his4-301(ACG) tif11Δ p5640 (sc LEU2 tif11-ΔG8)* | This study |
| PMY290 | *MATa ura3-52 trp1Δ63 leu2-3, leu2-112 his4-301(ACG) tif11Δ p5642 (sc LEU2 tif11-R13P)* | This study |
| PMY291 | *MATa ura3-52 trp1Δ63 leu2-3, leu2-112 his4-301(ACG) tif11Δ p5644 (sc LEU2 tif11-G15D)* | This study |
| PMY320 | *MATa ura3-52 trp1Δ63 leu2-3, leu2-112 his4-301(ACG) tif11Δ pDH469 (sc LEU2 tif11-K7A)* | This study |
| PMY321 | *MATa ura3-52 trp1Δ63 leu2-3, leu2-112 his4-301(ACG) tif11Δ pDH468 (sc LEU2 tif11-K7D)* | This study |
| PMY322 | *MATa ura3-52 trp1Δ63 leu2-3, leu2-112 his4-301(ACG) tif11Δ pDH481 (sc LEU2 tif11-ΔG8ΔG9)* | This study |
| PMY323 | *MATa ura3-52 trp1Δ63 leu2-3, leu2-112 his4-301(ACG) tif11Δ pDH471 (sc LEU2 tif11-K10A)* | This study |
| PMY324 | *MATa ura3-52 trp1Δ63 leu2-3, leu2-112 his4-301(ACG) tif11Δ pDH470 (sc LEU2 tif11-K10D)* | This study |
| PMY325 | *MATa ura3-52 trp1Δ63 leu2-3, leu2-112 his4-301(ACG) tif11Δ pDH473 (sc LEU2 tif11-R13A)* | This study |
| PMY326 | *MATa ura3-52 trp1Δ63 leu2-3, leu2-112 his4-301(ACG) tif11Δ pDH472 (sc LEU2 tif11-R13D)* | This study |
| PMY327 | *MATa ura3-52 trp1Δ63 leu2-3, leu2-112 his4-301(ACG) tif11Δ pDH475 (sc LEU2 tif11-R14A)* | This study |
| PMY329 | *MATa ura3-52 trp1Δ63 leu2-3, leu2-112 his4-301(ACG) tif11Δ pDH474 (sc LEU2 tif11-R14D)* | This study |
| PMY330 | *MATa ura3-52 trp1Δ63 leu2-3, leu2-112 his4-301(ACG) tif11Δ pDH478 (sc LEU2 tif11-K16A)* | This study |
| PMY332 | *MATa ura3-52 trp1Δ63 leu2-3, leu2-112 his4-301(ACG) tif11Δ pDH476 (sc LEU2 tif11-K16D)* | This study |
| PMY354 | *MATa ura3-52 trp1Δ63 leu2-3, leu2-112 his4-301(ACG) tif11Δ p3400 (hc LEU2 TIF11)* | This study |
| PMY355 | *MATa ura3-52 trp1Δ63 leu2-3, leu2-112 his4-301(ACG) tif11Δ pPMB167 (hc LEU2 tif11-K4D)* | This study |
| PMY357 | *MATa ura3-52 trp1Δ63 leu2-3, leu2-112 his4-301(ACG) tif11Δ pPMB168 (hc LEU2 tif11-ΔG8)* | This study |
| PMY358 | *MATa ura3-52 trp1Δ63 leu2-3, leu2-112 his4-301(ACG) tif11Δ pPMB169 (hc LEU2 tif11-ΔG8ΔG9)* | This study |
| PMY359 | *MATa ura3-52 trp1Δ63 leu2-3, leu2-112 his4-301(ACG) tif11Δ pPMB170 (hc LEU2 tif11- K10D)* | This study |
| PMY32 | *MATa ura3-52 leu2-3 leu2-112 trp1Δ−63 his4-301(ACG) sui1Δ::hisG pPMB02 (sc LEU2 sui1-K60E)* | (*Martin-Marcos et al., 2011*) |
| PMY293 | *MATa ura3-52 trp1Δ63 leu2-3, leu2-112 his4-301(ACG) tif11Δ p3390 (sc LEU2 TIF11) p4281 (sc TRP1 TIF5-G31R)* | This study |
| PMY295 | *MATa ura3-52 trp1Δ63 leu2-3, leu2-112 his4-301(ACG) tif11Δ p5633 (sc LEU2 tif11-K3E) p4281 (sc TRP1 TIF5-G31R)* | This study |
| PMY296 | *MATa ura3-52 trp1Δ63 leu2-3, leu2-112 his4-301(ACG) tif11Δ p5635 (sc LEU2 tif11-K4D) p4281 (sc TRP1 TIF5-G31R)* | This study |
| PMY297 | *MATa ura3-52 trp1Δ63 leu2-3, leu2-112 his4-301(ACG) tif11Δ p5638 (sc LEU2 tif11-T6D) p4281 (sc TRP1 TIF5-G31R)* | This study |
| PMY298 | *MATa ura3-52 trp1Δ63 leu2-3, leu2-112 his4-301(ACG) tif11Δ p5637 (sc LEU2 tif11-T6R) p4281 (sc TRP1 TIF5-G31R)* | This study |
| PMY300 | *MATa ura3-52 trp1Δ63 leu2-3, leu2-112 his4-301(ACG) tif11Δ p5640 (sc LEU2 tif11-ΔG8) p4281 (sc TRP1 TIF5-G31R)* | This study |
| PMY301 | *MATa ura3-52 trp1Δ63 leu2-3, leu2-112 his4-301(ACG) tif11Δ p5642 (sc LEU2 tif11-R13P) p4281 (sc TRP1 TIF5-G31R)* | This study |
| PMY302 | *MATa ura3-52 trp1Δ63 leu2-3, leu2-112 his4-301(ACG) tif11Δ p5644 (sc LEU2 tif11-G15D) p4281 (sc TRP1 TIF5-G31R)* | This study |
| PMY335 | *MATa ura3-52 trp1Δ63 leu2-3, leu2-112 his4-301(ACG) tif11Δ p3390 (sc LEU2 TIF11) p4280 (sc TRP1 SUI3-S264Y)* | This study |
| PMY310 | *MATa ura3-52 trp1Δ63 leu2-3, leu2-112 his4-301(ACG) tif11Δ p5633 (sc LEU2 tif11-K3E) p4280 (sc TRP1 SUI3-S264Y)* | This study |
| PMY311 | *MATa ura3-52 trp1Δ63 leu2-3, leu2-112 his4-301(ACG) tif11Δ p5635 (sc LEU2 tif11-K4D) p4280 (sc TRP1 SUI3-S264Y)* | This study |
| PMY312 | *MATa ura3-52 trp1Δ63 leu2-3, leu2-112 his4-301(ACG) tif11Δ p5638 (sc LEU2 tif11-T6D) p4280 (sc TRP1 SUI3-S264Y)* | This study |
| PMY313 | *MATa ura3-52 trp1Δ63 leu2-3, leu2-112 his4-301(ACG) tif11Δ p5637 (sc LEU2 tif11-T6R) p4280 (sc TRP1 SUI3-S264Y)* | This study |
| PMY315 | *MATa ura3-52 trp1Δ63 leu2-3, leu2-112 his4-301(ACG) tif11Δ p5640 (sc LEU2 tif11-ΔG8) p4280 (sc TRP1 SUI3-S264Y)* | This study |
| PMY316 | *MATa ura3-52 trp1Δ63 leu2-3, leu2-112 his4-301(ACG) tif11Δ p5642 (sc LEU2 tif11-R13P) p4280 (sc TRP1 SUI3-S264Y)* | This study |
| PMY317 | *MATa ura3-52 trp1Δ63 leu2-3, leu2-112 his4-301(ACG) tif11Δ p5644 (sc LEU2 tif11-G15D) p4280 (sc TRP1 SUI3-S264Y)* | This study |
| PMY339 | *MATa ura3-52 trp1Δ63 leu2-3, leu2-112 his4-301(ACG) tif11Δ pDH469 (sc LEU2 tif11-K7A) p4280 (sc TRP1 SUI3-S264Y)* | This study |
| PMY340 | *MATa ura3-52 trp1Δ63 leu2-3, leu2-112 his4-301(ACG) tif11Δ pDH468 (sc LEU2 tif11-K7D) p4280 (sc TRP1 SUI3-S264Y)* | This study |

*Table 2 continued on next page*

*Table 2 continued*

| Strain | Genotype | Source |
|--------|----------|--------|
| PMY341 | *MATa ura3-52 trp1Δ63 leu2-3, leu2-112 his4-301(ACG) tif11Δ pDH481 (sc LEU2 tif11-ΔG8ΔG9) p4280 (sc TRP1 SUI3-S264Y)* | This study |
| PMY342 | *MATa ura3-52 trp1Δ63 leu2-3, leu2-112 his4-301(ACG) tif11Δ pDH471 (sc LEU2 tif11-K10A) p4280 (sc TRP1 SUI3-S264Y)* | This study |
| PMY343 | *MATa ura3-52 trp1Δ63 leu2-3, leu2-112 his4-301(ACG) tif11Δ pDH470 (sc LEU2 tif11-K10D) p4280 (sc TRP1 SUI3-S264Y)* | This study |
| PMY344 | *MATa ura3-52 trp1Δ63 leu2-3, leu2-112 his4-301(ACG) tif11Δ pDH473 (sc LEU2 tif11-R13A) p4280 (sc TRP1 SUI3-S264Y)* | This study |
| PMY345 | *MATa ura3-52 trp1Δ63 leu2-3, leu2-112 his4-301(ACG) tif11Δ pDH472 (sc LEU2 tif11-R13D) p4280 (sc TRP1 SUI3-S264Y)* | This study |
| PMY346 | *MATa ura3-52 trp1Δ63 leu2-3, leu2-112 his4-301(ACG) tif11Δ pDH475 (sc LEU2 tif11-R14A) p4280 (sc TRP1 SUI3-S264Y)* | This study |
| PMY348 | *MATa ura3-52 trp1Δ63 leu2-3, leu2-112 his4-301(ACG) tif11Δ pDH474 (sc LEU2 tif11-R14D) p4280 (sc TRP1 SUI3-S264Y)* | This study |
| PMY349 | *MATa ura3-52 trp1Δ63 leu2-3, leu2-112 his4-301(ACG) tif11Δ pDH478 (sc LEU2 tif11-K16A) p4280 (sc TRP1 SUI3-S264Y)* | This study |
| PMY351 | *MATa ura3-52 trp1Δ63 leu2-3, leu2-112 his4-301(ACG) tif11Δ pDH476 (sc LEU2 tif11-K16D) p4280 (sc TRP1 SUI3-S264Y)* | This study |
| PMY337 | *MATa ura3-52 trp1Δ63 leu2-3, leu2-112 his4-301(ACG) tif11Δ p3390 (sc LEU2 TIF11) YCplac22 (sc TRP1)* | This study |
| PMY338 | *MATa ura3-52 trp1Δ63 leu2-3, leu2-112 his4-301(ACG) tif11Δ p3390 (sc LEU2 TIF11) YCplac22 (sc TRP1)* | This study |
| PMY360 | *MATa ura3-52 trp1Δ63 leu2-3, leu2-112 his4-301(ACG) tif11Δ p3400 (hc LEU2 TIF11) p4280 (sc TRP1 SUI3-S264Y)* | This study |
| PMY362 | *MATa ura3-52 trp1Δ63 leu2-3, leu2-112 his4-301(ACG) tif11Δ pPMB167 (hc LEU2 tif11-K4D) p4280 (sc TRP1 SUI3-S264Y)* | This study |
| PMY364 | *MATa ura3-52 trp1Δ63 leu2-3, leu2-112 his4-301(ACG) tif11Δ pPMB168 (hc LEU2 tif11-ΔG8) p4280 (sc TRP1 SUI3-S264Y)* | This study |
| PMY365 | *MATa ura3-52 trp1Δ63 leu2-3, leu2-112 his4-301(ACG) tif11Δ pPMB169 (hc LEU2 tif11-ΔG8ΔG9) p4280 (sc TRP1 SUI3-S264Y)* | This study |
| PMY366 | *MATa ura3-52 trp1Δ63 leu2-3, leu2-112 his4-301(ACG) tif11Δ pPMB170 (hc LEU2 tif11- K10D) p4280 (sc TRP1 SUI3-S264Y)* | This study |
| PMY361 | *MATa ura3-52 trp1Δ63 leu2-3, leu2-112 his4-301(ACG) tif11Δ p3400 (hc LEU2 TIF11) YCplac22 (sc TRP1)* | This study |
| GP3511 | *MATα ura3-52 leu2-3 leu2-112 ino1 sui2Δ gcn2Δ pep4::LEU2 < HIS4 lacZ,ura3−52 > pAV1089 (SUI2,SUI3,GCD11-HIS, URA3)* | (*Pavitt et al., 1998*) |
| H4560 | *MATα ura3-52 leu2-3 leu2-112 ino1 sui2Δ gcn2Δ pep4::leu2::natMX sui3Δ::kanMX < HIS4 lacZ,ura3−52 > p5321 (SUI2, SUI3-2,GCD11-HIS,LEU2)* | (*Martin-Marcos et al., 2014*) |
| YAS2488 | *MATa leu2-3,−112 his4-53a trp1 ura3-52 cup1::LEU2/PGK1 pG/MFA2 pG* | (*Algire et al., 2002*) |
| FZY010 | *MATa ura3-52 trp1Δ63 leu2-3, leu2-112 his4-301(ACG) tif11Δ p5642 (sc LEU2 tif11-R13P) YCplac22 (sc TRP1)* | This study |
| FZY011 | *MATa ura3-52 trp1Δ63 leu2-3, leu2-112 his4-301(ACG) tif11Δ p5642 (sc LEU2 tif11-R13P) YCplac22 (sc TRP1)* | This study |

DOI: https://doi.org/10.7554/eLife.31250.024

described (*Acker et al., 2007*). 40S subunits were purified as described previously from strain YAS2488 (*Acker et al., 2007*). Model mRNAs with sequences 5′-GGAA[UC]$_7$UAUG[CU]$_{10}$C-3′ and 5′-GGAA[UC]$_7$UUUG[CU]$_{10}$C-3′ were purchased from Thermo Scientific. Yeast tRNA$_i^{Met}$ was synthesized from a hammerhead fusion template using T7 RNA polymerase, charged with [$^{35}$S]-methionine, and used to prepare radiolabeled eIF2·GDPNP·[$^{35}$S]-Met-tRNA$_i$ ternary complexes ([$^{35}$S]-TC), all as previously described (*Acker et al., 2007*). Yeast Met-tRNA$_i^{Met}$ was purchased from tRNA Probes, LLC and used to prepare unlabeled TC in the same way. For eIF1A dissociation kinetics, the WT and mutant eIF1A proteins were labeled at their C-termini with Cys-Lys-ε-fluorescein dipeptide, using the Expressed Protein Ligation system as previously described (*Maag and Lorsch, 2003*).

## TC and eIF1A dissociation kinetics

TC dissociation rate constants ($k_{off}$) were measured by monitoring the amount of [$^{35}$S]-TC that remains bound to 40S·eIF1·eIF1A·mRNA (43S·mRNA) complexes over time, in the presence of excess unlabeled TC (chase), using a native gel shift assay to separate 40S-bound from unbound [$^{35}$S]-TC. 43S·mRNA complexes were preassembled for 2 hr at 26°C in reactions containing 40S subunits (20 nM), eIF1 (1 μM), eIF1A (WT or mutant variants, 1 μM), mRNA (10 μM), and [$^{35}$S]-TC (0.25 μM eIF2/0.1 mM GDPNP/1 nM [$^{35}$S]-Met-tRNA$_i$) in 60 μl of reaction buffer (30 mM Hepes-KOH (pH 7.4), 100 mM potassium acetate (pH 7.4), 3 mM magnesium acetate, and 2 mM dithiothreitol). To initiate each dissociation reaction, a 6 μl-aliquot of the preassembled 43S·mRNA complexes was

mixed with 3 µl of 3-fold concentrated unlabeled TC chase (comprised of 2 µM eIF2/0.3 mM GDPNP/0.9 µM Met-tRNA$_i$), to achieve in the final dissociation reaction a 300-fold excess of unlabeled TC (~300 nM) over labeled TC (~1 nM), based on the two different amounts of Met-tRNA$_i$ employed, and incubated for the prescribed period of time. A converging time course was employed so that all dissociation reactions are terminated simultaneously by the addition of native-gel dye and loaded directly on a running native gel. The fraction of [$^{35}$S]-Met-tRNA$_i$ remaining in 43S complexes at each time point was determined by quantifying the 40S-bound and unbound signals using a PhosphorImaging, normalized to the ratio observed at the earliest time-point, and the data were fit with a single exponential equation (*Kolitz et al., 2009*).

The kinetics of eIF1A dissociation were determined exactly as described earlier (*Saini et al., 2014*).

## Ribosome footprint profiling and RNA-Seq

Ribosome profiling was conducted essentially as described previously (*Sen et al., 2016*) as detailed below, on isogenic strains FZY010 and FZY011 (*tif11-R13P*), and PMY337 and PMY338 (WT *TIF11*), providing two biological replicates of each genotype, cultured in SC-Leu-Trp, except that cells were not treated with cycloheximide before harvesting, and cycloheximide was added to the lysis buffer at 5x the standard concentration. In addition, RNAse-trimmed ribosomes were purified by velocity sedimentation through sucrose gradients prior to extraction of mRNA; and Illumina Ribo-Zero Gold rRNA Removal Kit (Yeast) was employed on linker-ligated mRNA in lieu of poly(A) selection. Genes with less than 128 total mRNA reads, or less than 40 total RPF reads, in the four samples combined (two replicates of both WT and mutant strains) were excluded from the calculation of TE values.

### Generation, processing, and analysis of sequence libraries of ribosome protected footprints or total mRNA fragments

*tif11-R13P* (FZY010, FZY011) and WT (PMY337, PMY338) yeast strains growing exponentially in SC medium at 30°C were harvested by vacuum filtration at room temperature, without prior treatment with cycloheximide, and quick-frozen in liquid nitrogen. Cells were lysed in a freezer mill with lysis buffer (20 mM Tris [pH 8.0], 140 mM KCl, 1.5 mM MgCl$_2$, 1% Triton, 500 µg/mL cycloheximide). For ribosome footprint library preparation, 30 A$_{260}$ units of extract were treated with 450U of RNAse I (Ambion, #AM2295) for 1 hr at 25°C on a Thermomixer at 700 rpm, and 80S ribosomes were purified by sedimentation through a sucrose density gradient as described (*Guydosh and Green, 2014*). Ribosome-protected mRNA fragments (footprints) were purified using a miRNeasy Mini kit (Qiagen) per the vendor's instructions. Following size selection and dephosphorylation, a Universal miRNA cloning linker (New England Biolabs, #S1315S) was ligated to the 3' ends of footprints, followed by reverse transcription, circular ligation, rRNA subtraction, PCR amplification of the cDNA library, and DNA sequencing with an Illumina HiSeq system. For RNA-seq library preparation, total RNA was purified using miRNeasy Mini kit (Qiagen) from aliquots of the same extracts used for footprint library preparation, 5 µg total RNA was randomly fragmented at 70°C for 8 min in fragmentation reagent (Ambion #AM8740). Fragment size selection, library generation and sequencing were carried out as above, except Ribo-Zero Gold rRNA Removal Kit (Yeast) was employed to remove rRNAs after linker-ligation. Linker sequences were trimmed from Illumina reads and the trimmed fasta sequences were aligned to the *S. cerevisiae* ribosomal database using Bowtie (*Langmead et al., 2009*). The non-rRNA reads (unaligned reads) were then mapped to the *S. cerevisiae* genome using TopHat (*Trapnell et al., 2009*). Wiggle track normalization for viewing RPF or RNA reads in the IGV browser was conducted as follows. Wiggle files were produced from the alignment file, one each for genes on the Watson or Crick strand. The total reads on both strands were summed and a normalization factor *q* was calculated as 1000,000,000/(total reads on W + C strands). Wiggle files were then regenerated by multiplying all reads by the factor *q*, yielding the number of reads per 1000 million total reads (rpkm). uORFs with evidence of translation in WT and R13P cells were identified as follows. First, we employed the yassour-uorf program of (*Brar et al., 2012*) that identifies all potential uORFs within annotated 5'UTRs initiating with either AUG or a near-cognate codon and then quantifies the footprints in the +1 and −1 codons of all putative uORFs. A uORF was judged to be translated if the +1 to −1 footprint ratio exceeded four and the total footprint counts at +1 and −1 exceeded 15, and also if the reads in the zero frame are at least 50% of the reads in all three frames

(ie. -c15-r4-z0.5 in the relevant line of code). This analysis was conducted on multiple published and unpublished datasets summarized in Table S2, *Supplementary File 1*. After excluding uORFs shorter than three codons, we identified 564 AUG-initiated uORFs and 5497 near-cognate uORFs with evidence of translation in one or more experiments. In the second step, we validated ~51% and ~44% of the AUG uORFs and near-cognate uORFs, respectively, by employing a distinct uORF identification tool, RibORF (*Ji et al., 2015*), which is based on the criteria of 3-nt periodicity and uniformity of read distribution across uORF codons. Applying a moderately stringent probability of prediction of >0.5, RibORF confirmed that 291 AUG uORFs and 2429 near-cognate uORFs show evidence of translation in the datasets from which they were first identified by the yassour-uorf program. A bed file was generated containing the sequence coordinates of every uORF and combined with a bed file containing the coordinates of the 5′UTR, main CDS, and 3′UTR of each gene, and used to obtain footprint (FP) counts for 5′UTRs, uORFs, and main CDS in each strain examined, excluding the first and last nucleotide triplets of 5′UTRs, the first and last codons of uORFs, and the first 20 codons of CDS. mRNA read counts were determined for all codons of the main CDS. DESEQ (*Anders and Huber, 2010*) was employed for differential expression analysis of changes in TE, RPFs, or RRO values, and to impose cutoffs for minimum read numbers (as indicated in figure legends) and remove outliers.

For all notched box-plots, constructed using a web-based tool at http://shiny.chemgrid.org/box-plotr/, the upper and lower boxes contain the second and third quartiles and the band gives the median. If the notches in two plots do not overlap, there is roughly 95% confidence that their medians are different.

The AUG context adaptation index (context score) (*Miyasaka, 1999*) was calculated as $A_{UG}CAI = (w_{-6} \times w_{-5} \times w_{-4} \times w_{-3} \times w_{-2} \times w_{-1} \times w_{+1} \times w_{+2} \times w_{+3})^{1/9}$ where $w_i$ is the fractional occurrence of that particular base, normalized to the most prevalent base, present in the $i^{th}$ position of the context among the ~270 most highly expressed yeast genes, taken from the matrix of frequencies and relative adaptiveness (w) of the nucleotide in the AUG context of this group of ~270 reference genes (*Zur and Tuller, 2013*). The context scores range from ~0.16 (poorest) to ~0.97 (best) among all yeast genes.

## Accession numbers

Sequencing data from this study have been submitted to the NCBI Gene Expression Omnibus (GEO; http://www.ncbi.nlm.nih.gov/geo/) and the accession numbers are listed in the Additional Files under Major datasets.

## Acknowledgements

We thank Nicholas Ingolia, Nicholas Guydosh, and David Young for protocols and helpful discussions about ribosome profiling data analysis, Swati Gaikwad for help in analysis of AUG context scores and Shardul Kulkarni for sharing ribosome profiling data prior to publication. This work was supported in part by the Intramural Research Program of the National Institutes of Health. PM-M was financed in part by the Spanish Ministry of Economy, Industry and Competitiveness (MINECO) and European FEDER funds, through Project RTC2015-4391-2 awarded to MT.

## Additional information

### Competing interests

Alan G Hinnebusch: Reviewing editor, *eLife*. The other authors declare that no competing interests exist.

### Funding

| Funder | Grant reference number | Author |
| --- | --- | --- |
| National Institutes of Health | Intramural Research Program | Alan G Hinnebusch |

The funders had no role in study design, data collection and interpretation, or the decision to submit the work for publication.

## Author contributions

Pilar Martin-Marcos, Conceptualization, Formal analysis, Validation, Investigation, Visualization, Methodology, Writing—original draft, Writing—review and editing; Fujun Zhou, Data curation, Formal analysis, Validation, Investigation, Visualization, Writing—original draft, Writing—review and editing; Charm Karunasiri, Investigation; Fan Zhang, Jagpreet Nanda, Formal analysis, Investigation, Writing—original draft, Writing—review and editing; Jinsheng Dong, Investigation, Writing—original draft; Shardul D Kulkarni , Alan G Hinnebusch, Investigation, Formal analysis; Neelam Dabas Sen, Resources, Writing—original draft, Writing—review and editing; Mercedes Tamame, Investigation, Writing—original draft, Writing—review and editing; Michael Zeschnigk, Supervision, Writing—original draft, Writing—review and editing; Jon R Lorsch, Conceptualization, Resources, Funding acquisition, Validation, Visualization, Methodology, Writing—original draft, Project administration, Writing—review and editing

## Author ORCIDs

Pilar Martin-Marcos (iD) https://orcid.org/0000-0001-8897-099X
Jon R Lorsch (iD) https://orcid.org/0000-0002-4521-4999
Alan G Hinnebusch (iD) http://orcid.org/0000-0002-1627-8395

## Decision letter and Author response

Decision letter https://doi.org/10.7554/eLife.31250.105
Author response https://doi.org/10.7554/eLife.31250.106

## Additional files

### Supplementary files

• Supplementary file 1. Supplementary Tables. Table S1: Oligonucleotide primers employed for *TIF11* mutagenesis in this study Table S2: Ribosome profiling datasets used for uORF identification
DOI: https://doi.org/10.7554/eLife.31250.025

• Supplementary file 2. Excel file containing results and analyses from ribosome footprint profiling of WT and *tif11-R13P* cells. Spreadsheet 1, '*CDS*_all Expr', tabulates $\log_2$ values of the following parameters for the 5037 expressed genes listed in columns A-B for WT and *tif11-R13P* cells: Ribosome footprint sequencing reads (RPF_WT and RPF_*R13P*); mRNA sequencing reads (mRNA_WT and mRNA_*R13P*); the ratios RPF_*R13P*/RPF_WT (ΔRPF_ *R13P*) and mRNA_*R13P*/mRNA_WT (ΔmRNA_*R13P*); and the ratio ΔRPF_*R13P*/ΔmRNA_*R13P* (ΔTE_*R13P*). Spreadsheet 2, 'Context_-score', contains a subset of genes in Spreadsheet 1 (4280 genes) with 5'UTR length >5 nt examined for additional parameters: 5'UTR length, sequences between positions −6 and +6, and the context scores. Spreadsheet 3, 'AUG_uORFs_identified', contains all 564 AUG uORFs identified using the yassour-uorf program from multiple datasets listed in Table S2, listing uORF chromosome coordinates, distances of the uORF AUG from the 5' end of the mRNA and the main CDS start codon, uORF sequence contexts between the −3 and +4 positions and the context scores (NA, for uORF 5' UTR <3 nt). Spreadsheet 4, '*uORF_Expr*', tabulates $\log_2$ values of the following parameters for the 97 expressed uAUG uORFs listed in column A for WT and *tif11-R13P* cells: Ribosome footprint sequencing reads on CDS (RPF_CDS_WT and RPF_CDS_*R13P*); Ribosome footprint sequencing reads on uORFs (RPF_uORF_WT and RPF_uORF_*R13P*); the ratios RPF_CDS_*R13P*/RPF_CDS_WT (ΔRPF_CDS_ *R13P*) and RPF_uORF_*R13P*/RPF_uORF_WT (ΔRPF_uORF_*R13P*); Relative Ribosome Occupancy (RRO) for WT (the ratio of RPF_uORF_WT/RPF_CDS_WT, RRO_WT) and *R13P* (the ratio of RPF_uORF_*R13P*/RPF_CDS_*R13P*, RRO_R13P) and the ratio RRO_*R13P*/RRO_*WT* (RRO_*R13P*); Spreadsheet 5, PARS score analysis of 5' UTRs of 2642 genes, conducted as described in (*Sen et al., 2016*) and their context scores as listed in Spreadsheet 2. Spreadsheet 6 '*CDS*_RPF_Change', tabulates $\log_2$ values of the ratio RPF_*R13P*/RPF_WT (ΔRPF_*R13P* (log2)) and adjusted p-value (padj) for the 5083 expressed genes detected by the DESEQ2 package listed in columns A-B

DOI: https://doi.org/10.7554/eLife.31250.026
• Transparent reporting form
DOI: https://doi.org/10.7554/eLife.31250.027

## Major datasets

The following datasets were generated:

| Author(s) | Year | Dataset title | Dataset URL | Database, license, and accessibility information |
|---|---|---|---|---|
| Pilar Martin-Marcos, Fujun Zhou, Charm Karunasiri, Fan Zhang, Jinsheng Dong, Jagpreet Nanda, Shardul D Kulkarni, Neelam Dabas Sen, Mercedes Tamame, Michael Zeschnigk, Jon R Lorsch, Alan G Hinnebusch | 2017 | eIF1A residues implicated in cancer stabilize translation preinitiation complexes and favor suboptimal initiation sites in yeast | https://www.ncbi.nlm.nih.gov/geo/query/acc.cgi?acc=GSE108334 | Publicly available at the NCBI Gene Expression Omnibus (accession no. GSE108334) |
| Pilar Martin-Marcos, Fujun Zhou, Charm Karunasiri, Fan Zhang, Jinsheng Dong, Jagpreet Nanda, Shardul D Kulkarni, Neelam Dabas Sen, Mercedes Tamame, Michael Zeschnigk, Jon R Lorsch, Alan G Hinnebusch | 2017 | ribo_WT(for SUI1-L96P)_with_CHX_1 | https://www.ncbi.nlm.nih.gov/geo/query/acc.cgi?acc=GSM2895470 | Publicly available at the NCBI Gene Expression Omnibus (accession no. GSM2895470) |
| Pilar Martin-Marcos, Fujun Zhou, Charm Karunasiri, Fan Zhang, Jinsheng Dong, Jagpreet Nanda, Shardul D Kulkarni, Neelam Dabas Sen, Mercedes Tamame, Michael Zeschnigk, Jon R Lorsch, Alan G Hinnebusch | 2017 | ribo_WT(for SUI1-L96P)_with_CHX_2 | https://www.ncbi.nlm.nih.gov/geo/query/acc.cgi?acc=GSM2895471 | Publicly available at the NCBI Gene Expression Omnibus (accession no. GSM2895471) |
| Pilar Martin-Marcos, Fujun Zhou, Charm Karunasiri, Fan Zhang, Jinsheng Dong, Jagpreet Nanda, Shardul D Kulkarni, Neelam Dabas Sen, Mercedes Tamame, Michael Zeschnigk, Jon R Lorsch, Alan G Hinnebusch | 2017 | ribo_SUI1-L96P_w/o_CHX_1 | https://www.ncbi.nlm.nih.gov/geo/query/acc.cgi?acc=GSM2895472 | Publicly available at the NCBI Gene Expression Omnibus (accession no. GSM2895472) |

| | | | | |
|---|---|---|---|---|
| Pilar Martin-Marcos, Fujun Zhou, Charm Karunasiri, Fan Zhang, Jinsheng Dong, Jagpreet Nanda, Shardul D Kulkarni, Neelam Dabas Sen, Mercedes Tamame, Michael Zeschnigk, Jon R Lorsch, Alan G Hinnebusch | 2017 | ribo_SUI1-L96P_w/o_CHX_2 | https://www.ncbi.nlm.nih.gov/geo/query/acc.cgi?acc=GSM2895473 | Publicly available at the NCBI Gene Expression Omnibus (accession no. GSM2895473) |
| Pilar Martin-Marcos, Fujun Zhou, Charm Karunasiri, Fan Zhang, Jinsheng Dong, Jagpreet Nanda, Shardul D Kulkarni, Neelam Dabas Sen, Mercedes Tamame, Michael Zeschnigk, Jon R Lorsch, Alan G Hinnebusch | 2017 | ribo_SUI1-L96P_with_CHX_1 | https://www.ncbi.nlm.nih.gov/geo/query/acc.cgi?acc=GSM2895468 | Publicly available at the NCBI Gene Expression Omnibus (accession no. GSM2895468) |
| Pilar Martin-Marcos, Fujun Zhou, Charm Karunasiri, Fan Zhang, Jinsheng Dong, Jagpreet Nanda, Shardul D Kulkarni, Neelam Dabas Sen, Mercedes Tamame, Michael Zeschnigk, Jon R Lorsch, Alan G Hinnebusch | 2017 | ribo_SUI1-L96P_with_CHX_2 | https://www.ncbi.nlm.nih.gov/geo/query/acc.cgi?acc=GSM2895469 | Publicly available at the NCBI Gene Expression Omnibus (accession no. GSM2895469) |
| Pilar Martin-Marcos, Fujun Zhou, Charm Karunasiri, Fan Zhang, Jinsheng Dong, Jagpreet Nanda, Shardul D Kulkarni, Neelam Dabas Sen, Mercedes Tamame, Michael Zeschnigk, Jon R Lorsch, Alan G Hinnebusch | 2017 | ribo_WT(for SUI1-T15A)_1 | https://www.ncbi.nlm.nih.gov/geo/query/acc.cgi?acc=GSM2895458 | Publicly available at the NCBI Gene Expression Omnibus (accession no. GSM2895458) |
| Pilar Martin-Marcos, Fujun Zhou, Charm Karunasiri, Fan Zhang, Jinsheng Dong, Jagpreet Nanda, Shardul D Kulkarni, Neelam Dabas Sen, Mercedes Tamame, Michael Zeschnigk, Jon R Lorsch, Alan G Hinnebusch | 2017 | ribo_WT(for SUI1-T15A)_2 | https://www.ncbi.nlm.nih.gov/geo/query/acc.cgi?acc=GSM2895459 | Publicly available at the NCBI Gene Expression Omnibus (accession no. GSM2895459) |

| | | | | |
|---|---|---|---|---|
| Pilar Martin-Marcos, Fujun Zhou, Charm Karunasiri, Fan Zhang, Jinsheng Dong, Jagpreet Nanda, Shardul D Kulkarni, Neelam Dabas Sen, Mercedes Tamame, Michael Zeschnigk, Jon R Lorsch, Alan G Hinnebusch | 2017 | ribo_SUI1-T15A_1 | https://www.ncbi.nlm.nih.gov/geo/query/acc.cgi?acc=GSM2895456 | Publicly available at the NCBI Gene Expression Omnibus (accession no. GSM2895456) |
| Pilar Martin-Marcos, Fujun Zhou, Charm Karunasiri, Fan Zhang, Jinsheng Dong, Jagpreet Nanda, Shardul D Kulkarni, Neelam Dabas Sen, Mercedes Tamame, Michael Zeschnigk, Jon R Lorsch, Alan G Hinnebusch | 2017 | ribo_SUI-T15A_2 | https://www.ncbi.nlm.nih.gov/geo/query/acc.cgi?acc=GSM2895457 | Publicly available at the NCBI Gene Expression Omnibus (accession no. GSM2895457) |
| Pilar Martin-Marcos, Fujun Zhou, Charm Karunasiri, Fan Zhang, Jinsheng Dong, Jagpreet Nanda, Shardul D Kulkarni, Mercedes Tamame, Michael Zeschnigk, Jon R Lorsch, Alan G Hinnebusch | 2017 | ribo_SUI3-2(for SUI1_T15A)_1 | https://www.ncbi.nlm.nih.gov/geo/query/acc.cgi?acc=GSM2895460 | Publicly available at the NCBI Gene Expression Omnibus (accession no. GSM2895460) |
| Pilar Martin-Marcos, Fujun Zhou, Charm Karunasiri, Fan Zhang, Jinsheng Dong, Jagpreet Nanda, Shardul D Kulkarni, Neelam Dabas Sen, Mercedes Tamame, Michael Zeschnigk, Jon R Lorsch, Alan G Hinnebusch | 2017 | ribo_SUI3-2(for SUI1_T15A)_2 | https://www.ncbi.nlm.nih.gov/geo/query/acc.cgi?acc=GSM2895461 | Publicly available at the NCBI Gene Expression Omnibus (accession no. GSM2895461) |
| Pilar Martin-Marcos, Fujun Zhou, Charm Karunasiri, Fan Zhang, Jinsheng Dong, Jagpreet Nanda, Shardul D Kulkarni, Neelam Dabas Sen, Mercedes Tamame, Michael Zeschnigk, Jon R Lorsch, Alan G Hinnebusch | 2017 | ribo_SUI3-2/SUI-T15A_1 | https://www.ncbi.nlm.nih.gov/geo/query/acc.cgi?acc=GSM2895462 | Publicly available at the NCBI Gene Expression Omnibus (accession no. GSM2895462) |
| Pilar Martin-Marcos, Fujun Zhou, Charm Karunasiri, Fan Zhang, Jinsheng Dong, Jagpreet Nanda, Shardul D Kulkarni, Neelam Dabas Sen, Mercedes Tamame, Michael Zeschnigk, Jon R Lorsch, Alan G Hinnebusch | 2017 | ribo_SUI3-2/SUI-T15A_2 | https://www.ncbi.nlm.nih.gov/geo/query/acc.cgi?acc=GSM2895463 | Publicly available at the NCBI Gene Expression Omnibus (accession no. GSM2895463) |

| | | | | |
|---|---|---|---|---|
| Pilar Martin-Marcos, Fujun Zhou, Charm Karunasiri, Fan Zhang, Jinsheng Dong, Jagpreet Nanda, Shardul D Kulkarni, Neelam Dabas Sen, Mercedes Tamame, Michael Zeschnigk, Jon R Lorsch, Alan G Hinnebusch | 2017 | ribo_20_WT_with_CHX_1 | https://www.ncbi.nlm.nih.gov/geo/query/acc.cgi?acc=GSM2895476 | Publicly available at the NCBI Gene Expression Omnibus (accession no. GSM2895476) |
| Pilar Martin-Marcos, Fujun Zhou, Charm Karunasiri, Fan Zhang, Jinsheng Dong, Jagpreet Nanda, Shardul D Kulkarni, Neelam Dabas Sen, Mercedes Tamame, Michael Zeschnigk, Jon R Lorsch, Alan G Hinnebusch | 2017 | ribo_WT(for TIF11_R13P)_1 | https://www.ncbi.nlm.nih.gov/geo/query/acc.cgi?acc=GSM2895450 | Publicly available at the NCBI Gene Expression Omnibus (accession no. GSM2895450) |
| Pilar Martin-Marcos, Fujun Zhou, Charm Karunasiri, Fan Zhang, Jinsheng Dong, Jagpreet Nanda, Shardul D Kulkarni, Neelam Dabas Sen, Mercedes Tamame, Michael Zeschnigk, Jon R Lorsch, Alan G Hinnebusch | 2017 | ribo_WT(for TIF11_R13P)_2 | https://www.ncbi.nlm.nih.gov/geo/query/acc.cgi?acc=GSM2895451 | Publicly available at the NCBI Gene Expression Omnibus (accession no. GSM2895451) |
| Pilar Martin-Marcos, Fujun Zhou, Charm Karunasiri, Fan Zhang, Jinsheng Dong, Jagpreet Nanda, Shardul D Kulkarni, Neelam Dabas Sen, Mercedes Tamame, Michael Zeschnigk, Jon R Lorsch, Alan G Hinnebusch | 2017 | ribo_TIF11_R13P_1 | https://www.ncbi.nlm.nih.gov/geo/query/acc.cgi?acc=GSM2895448 | Publicly available at the NCBI Gene Expression Omnibus (accession no. GSM2895448) |
| Pilar Martin-Marcos, Charm Karunasiri, Fujun Zhou, Fan Zhang, Jinsheng Dong, Jagpreet Nanda, Shardul D Kulkarni, Neelam Dabas Sen, Mercedes Tamame, Michael Zeschnigk, Jon R Lorsch, Alan G Hinnebusch | 2017 | ribo_SUI3-2(for TIF11-R13P)_1 | https://www.ncbi.nlm.nih.gov/geo/query/acc.cgi?acc=GSM2895452 | Publicly available at the NCBI Gene Expression Omnibus (accession no. GSM2895452) |

| | | | | |
|---|---|---|---|---|
| Pilar Martin-Marcos, Fujun Zhou, Charm Karunasiri, Fan Zhang, Jinsheng Dong, Jagpreet Nanda, Shardul D Kulkarni, Neelam Dabas Sen, Mercedes Tamame, Michael Zeschnigk, Jon R Lorsch, Alan G Hinnebusch | 2017 | ribo_SUI3-2(for TIF11-R13P)_2 | https://www.ncbi.nlm. nih.gov/geo/query/acc. cgi?acc=GSM2895453 | Publicly available at the NCBI Gene Expression Omnibus (accession no. GSM2895453) |
| Pilar Martin-Marcos, Fujun Zhou, Charm Karunasiri, Fan Zhang, Jinsheng Dong, Jagpreet Nanda, Shardul D Kulkarni, Neelam Dabas Sen, Mercedes Tamame, Michael Zeschnigk, Jon R Lorsch, Alan G Hinnebusch | 2017 | ribo_SUI3-2/TIF11_R13P_1 | https://www.ncbi.nlm. nih.gov/geo/query/acc. cgi?acc=GSM2895454 | Publicly available at the NCBI Gene Expression Omnibus (accession no. GSM2895454) |
| Pilar Martin-Marcos, Fujun Zhou, Charm Karunasiri, Fan Zhang, Jinsheng Dong, Jagpreet Nanda, Shardul D Kulkarni, Neelam Dabas Sen, Mercedes Tamame, Michael Zeschnigk, Jon R Lorsch, Alan G Hinnebusch | 2017 | ribo_SUI3-2/TIF11_R13P_2 | https://www.ncbi.nlm. nih.gov/geo/query/acc. cgi?acc=GSM2895455 | Publicly available at the NCBI Gene Expression Omnibus (accession no. GSM2895455) |
| Pilar Martin-Marcos, Fujun Zhou, Charm Karunasiri, Fan Zhang, Jinsheng Dong, Jagpreet Nanda, Shardul D Kulkarni, Neelam Dabas Sen, Mercedes Tamame, Michael Zeschnigk, Jon R Lorsch, Alan G Hinnebusch | 2017 | ribo_WT(for SUI1-L96P)_w/o_CHX_1 | https://www.ncbi.nlm. nih.gov/geo/query/acc. cgi?acc=GSM2895474 | Publicly available at the NCBI Gene Expression Omnibus (accession no. GSM2895474) |
| Pilar Martin-Marcos, Fujun Zhou, Charm Karunasiri, Jinsheng Dong, Jagpreet Nanda, Shardul D Kulkarni, Neelam Dabas Sen, Mercedes Tamame, Michael Zeschnigk, Jon R Lorsch, Alan G Hinnebusch, Fan Zhang | 2017 | ribo_WT(for SUI1-L96P)_w/o_CHX_2 | https://www.ncbi.nlm. nih.gov/geo/query/acc. cgi?acc=GSM2895475 | Publicly available at the NCBI Gene Expression Omnibus (accession no. GSM2895475 |

| | | | | |
|---|---|---|---|---|
| Pilar Martin-Marcos, Fujun Zhou, Charm Karunasiri, Fan Zhang, Jinsheng Dong, Jagpreet Nanda, Shardul D Kulkarni, Neelam Dabas Sen, Mercedes Tamame, Michael Zeschnigk, Jon R Lorsch, Alan G Hinnebusch | 2017 | ribo_37_WT_w/o_CHX_2 | https://www.ncbi.nlm.nih.gov/geo/query/acc.cgi?acc=GSM2895491 | Publicly available at the NCBI Gene Expression Omnibus (accession no. GSM2895491) |
| Pilar Martin-Marcos, Fujun Zhou, Charm Karunasiri, Fan Zhang, Jinsheng Dong, Jagpreet Nanda, Shardul D Kulkarni, Neelam Dabas Sen, Mercedes Tamame, Michael Zeschnigk, Jon R Lorsch, Alan G Hinnebusch | 2017 | ribo_WT(for tif3)_1 | https://www.ncbi.nlm.nih.gov/geo/query/acc.cgi?acc=GSM2895466 | Publicly available at the NCBI Gene Expression Omnibus (accession no. GSM2895466) |
| Pilar Martin-Marcos, Fujun Zhou, Charm Karunasiri, Fan Zhang, Jinsheng Dong, Jagpreet Nanda, Shardul D Kulkarni, Neelam Dabas Sen, Mercedes Tamame, Michael Zeschnigk, Jon R Lorsch, Alan G Hinnebusch | 2017 | ribo_WT(for tif3)_2 | https://www.ncbi.nlm.nih.gov/geo/query/acc.cgi?acc=GSM2895467 | Publicly available at the NCBI Gene Expression Omnibus (accession no. GSM2895467) |
| Pilar Martin-Marcos, Fujun Zhou, Charm Karunasiri, Fan Zhang, Jinsheng Dong, Jagpreet Nanda, Shardul D Kulkarni, Neelam Dabas Sen, Mercedes Tamame, Michael Zeschnigk, Jon R Lorsch, Alan G Hinnebusch | 2017 | ribo_tif3△_1 | https://www.ncbi.nlm.nih.gov/geo/query/acc.cgi?acc=GSM2895464 | Publicly available at the NCBI Gene Expression Omnibus (accession no. GSM2895464) |
| Pilar Martin-Marcos, Fujun Zhou, Charm Karunasiri, Fan Zhang, Jinsheng Dong, Jagpreet Nanda, Shardul D Kulkarni, Neelam Dabas Sen, Mercedes Tamame, Michael Zeschnigk, Jon R Lorsch, Alan G Hinnebusch | 2017 | ribo_tif3△_2 | https://www.ncbi.nlm.nih.gov/geo/query/acc.cgi?acc=GSM2895465 | Publicly available at the NCBI Gene Expression Omnibus (accession no. GSM2895465) |

| Pilar Martin-Marcos, Fujun Zhou, Charm Karunasiri, Fan Zhang, Jinsheng Dong, Jagpreet Nanda, Shardul D Kulkarni, Neelam Dabas Sen, Mercedes Tamame, Michael Zeschnigk, Jon R Lorsch, Alan G Hinnebusch | 2017 | ribo_TIF11_R13P_2 | https://www.ncbi.nlm.nih.gov/geo/query/acc.cgi?acc=GSM2895449 | Publicly available at the NCBI Gene Expression Omnibus (accession no. GSM2895449) |

The following previously published datasets were used:

| Author(s) | Year | Dataset title | Dataset URL | Database, license, and accessibility information |
|---|---|---|---|---|
| Gerashchenko M, Gladyshev V | 2014 | Translation Inhibitors Cause Abnormalities in Ribosome Profiling Experiments | https://www.ncbi.nlm.nih.gov/geo/query/acc.cgi?acc=GSE59573 | Publicly available at the NCBI Gene Expression Omnibus (accession no: GSE59573) |
| Guydosh NR, Green R | 2013 | Ribosome profiling study of dom34 and hbs1 knockout strains using short (16-nt) and long (28-nt) monosome-protected footprints and disome-protected footprints | https://www.ncbi.nlm.nih.gov/geo/query/acc.cgi?acc=GSE52968 | Publicly available at the NCBI Gene Expression Omnibus (accession no: GSE52968) |
| Sen ND, Zhou F, Ingolia NT, Hinnebusch AG | 2015 | Genome-wide analysis of translational efficiency reveals distinct but overlapping functions of yeast DEAD-box RNA helicases Ded1 and eIF4A | https://www.ncbi.nlm.nih.gov/geo/query/acc.cgi?acc=GSE66411 | Publicly available at the NCBI Gene Expression Omnibus (accession no: GSE66411) |
| Sen ND, Zhou F, Harris MS, Ingolia NT, Hinnebusch AG | 2016 | eIF4B preferentially stimulates translation of long mRNAs with structured 5'UTRs and low closed-loop potential but weak dependence on eIF4G | https://www.ncbi.nlm.nih.gov/geo/query/acc.cgi?acc=GSE81966 | Publicly available at the NCBI Gene Expression Omnibus (accession no: GSE81966) |
| Young DJ, Guydosh NR, Zhang F, Hinnebusch AG, Green R | 2015 | Ribosome profiling study of rli1 depletion strain | https://www.ncbi.nlm.nih.gov/geo/query/acc.cgi?acc=GSE69414 | Publicly available at the NCBI Gene Expression Omnibus (accession no: GSE69414) |
| Kertesz M, Wan Y, Mazor E, Rinn JL, Nutter RC, Chang HY, Segal E | 2010 | Genome-wide Measurement of RNA Secondary Structure in Yeast | https://www.ncbi.nlm.nih.gov/geo/query/acc.cgi?acc=GSE22393 | Publicly available at the NCBI Gene Expression Omnibus (accession no: GSE22393) |
| Pelechano V, Wei W, Steinmetz LM | 2013 | Saccharomyces cerevisiae Transcript Isoform mapping | https://www.ncbi.nlm.nih.gov/geo/query/acc.cgi?acc=GSE39128 | Publicly available at the NCBI Gene Expression Omnibus (accession no: GSE39128) |

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
