## [Decision Letter]

Thank you for submitting your article "eIF1A residues implicated in cancer stabilize translation preinitiation complexes and favor suboptimal initiation sites" for consideration by *eLife*. Your article has been favorably evaluated by James Manley (Senior Editor) and three reviewers, one of whom, Nahum Sonenberg (Reviewer #1), is a member of our Board of Reviewing Editors. The following individual involved in review of your submission has agreed to reveal their identity: Matt Sachs (Reviewer #3).

The reviewers have discussed the reviews with one another and the Reviewing Editor has drafted this decision to help you prepare a revised submission.

Summary:

The authors present a study that provides an extensive analysis of initiation codon selection in response to N-terminal mutants of yeast eIF1A. The study primarily focuses on cancer-associated eIF1A-NTT mutants. Cancer-associated mutants are introduced into the yeast eIF1A protein together with some synthetic yeast mutants in the same region. Mutants confer a hyper-accuracy phenotype that is likely attributable to destabilization of a closed/Pin state of the scanning 48S PIC. The most potent mutant R13P confers altered start site selection genome-wide in yeast, which is consistent with the genetic experiments and in vitro kinetics.

Previous work from Hinnebusch and coworkers has shown that specific residues in eIF1, eIF1A and eIF2β function in the discrimination of poor context AUG recognition and non-AUG codons (Martin-Marcos et al. MCB 2011 Dec; 31(23): 4814-4831; Saini et al. 2010 Genes Dev. 24: 97-110). The current study now provides far more detail into how eIF1A plays a role in start site selection. Interestingly, the finding that essentially all cancer-associated mutations of eIF1A, when made in yeast eIF1A, confer a consistent hyper-accuracy phenotype, is very striking. Overall, the experiments are well undertaken and the data are appropriately interpreted.

Importantly, the authors' findings indicate that the increased stringency of selection of AUG codons in the optimal Kozak context favors oncogenic transformation. The authors offer the reasonable prediction "If one or more tumor suppressor genes contains such a poor initiation site, the μm substitutions can be expected to increase its relative translation rate and thereby impair one or more control mechanisms governing cell proliferation"

Essential revisions:

1) One concern of the study is whether the cancer-associated mutants in eIF1A are applicable to human eIF1A function and cancer. The eIF1A-NTT is highly conserved between two species, but the Kozak consensus and dependence on it are generally thought to vary between yeast and humans. The study, therefore, provides new insight into how the eIF1A-NTT maintains the fidelity of start codon selection, but the possible role of the cancer-associated residues in human eIF1A in mammalian start site selection is not determined. Ideally, this should be addressed directly, but at a minimum the end of the title should be altered to include "in yeast" to better describe the study.

2) They must have become aware of a very relevant paper Johnson et al. PLoS One. 2017 Jun 8;12(6):e0178189). The authors did RNAseq of polysome fractions prepared from actual μm cancer cells harboring the eIF1A-NTT mutation. They identified genes whose translational efficiency was affected by the knockdown of EIF1AX. Could the authors look for relevant information regarding mRNAs from the list? The paper should be referenced and its implications discussed.

3) Figure 5 convincingly demonstrates that the R13P and K16D mutations in eIF1A destabilize the closed/Pin conformation of the 48S PIC at UUG codons in vitro. Could they extend these results by showing a decreased UUG/AUG start codon usage in a cell-free translation system from eIF1A-mutant cells as compared to wildtype?

---

## [Author Response]

Essential revisions:1) One concern of the study is whether the cancer-associated mutants in eIF1A are applicable to human eIF1A function and cancer. The eIF1A-NTT is highly conserved between two species, but the Kozak consensus and dependence on it are generally thought to vary between yeast and humans. The study, therefore, provides new insight into how the eIF1A-NTT maintains the fidelity of start codon selection, but the possible role of the cancer-associated residues in human eIF1A in mammalian start site selection is not determined. Ideally, this should be addressed directly, but at a minimum the end of the title should be altered to include "in yeast" to better describe the study.

Ribosome profiling of human cells altered by gene editing to express the R13P tumor mutation is being planned, but is beyond the scope of this paper. Hence, we have altered the title to include “in yeast”.

2) They must have become aware of a very relevant paper Johnson et al. PLoS One. 2017 Jun 8;12(6):e0178189). The authors did RNAseq of polysome fractions prepared from actual μm cancer cells harboring the eIF1A-NTT mutation. They identified genes whose translational efficiency was affected by the knockdown of EIF1AX. Could the authors look for relevant information regarding mRNAs from the list? The paper should be referenced and its implications discussed.

We have added new material to the Discussion to address these findings, stating: “A recent study on μm tumor cell lines expressing either WT or the G6D variant of EIF1AX provided evidence that the G6D substitution reduces the critical function of EIF1AX in bulk translation initiation. […] Thus, it is possible that the reduction in RPG expression in G6D tumor cells is a response to reduced bulk translation and cell growth; and it will be interesting to determine whether the R13P substitution in EIF1X increases discrimination against AUGs in poor context in human cells.”

3) Figure 5 convincingly demonstrates that the R13P and K16D mutations in eIF1A destabilize the closed/Pin conformation of the 48S PIC at UUG codons in vitro. Could they extend these results by showing a decreased UUG/AUG start codon usage in a cell-free translation system from eIF1A-mutant cells as compared to wildtype?

While we were not able to comply with this specific request, we were able to add pertinent new results that confirm the effects of both the R13P and K16D substitutions in reducing the UUG:AUG ratio in yeast cells using an independent assay based on luciferase reporters harboring UUG or AUG start codons. The new results are shown in the Figure 3—figure supplement 1, and completely support the findings obtained using the *HIS4-lacZ* UUG and AUG reporters. Given the complete agreement between these two orthogonal in vivo assays and the effects of these substitutions in destabilizing the closed/Pin conformation of reconstituted 48S PIC at UUG codons in vitro, we hope the reviewers will agree that the evidence is very strong that these eIF1A substitutions increase discrimination against a near-cognate start codon.